# Spatial Expansion, Planning, and Their Influences on the Urban Landscape of Christian Churches in Canton (1582–1732 and 1844–1911)

**Yonggu Li**

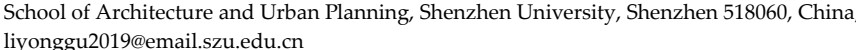

School of Architecture and Urban Planning, Shenzhen University, Shenzhen 518060, China;
liyonggu2019@email.szu.edu.cn

**Abstract:** Canton (present-day Guangzhou, China) has a long history as a trading port and serves as a window for studying the history of Sino-Western cultural exchanges. Canton was a city built under Confucian orders, leading to significant differences (when compared to Christian cities) in urban functional zoning, layout, urban landscape, and methods for shaping spatial order. Therefore, the churches constructed by Christian missionary societies in Canton merit particular attention in missionary history research and urban planning history. Based on local gazetteers, historical maps, export paintings, Western travelogues, and archives at that time, from a cultural landscape perspective, this article compares and analyzes the spatial expansion of Christian churches and their influences on the urban landscape in Canton in two stages. In the late Ming and early Qing dynasties, the spatial layout of the churches indicated an active integration into Canton City. After the Opium War, churches were not only used for religious purposes but also served as symbols asserting the presence of Christians and Western powers (which made the situation more complicated). Missionary societies attracted believers through the construction of public facilities, building Christian communities centered around churches, thereby competing with authorities for spatial power and influencing the urban functional system and spatial layout controlled by the authorities. Comparatively, the Roman Catholic Cathedral has profoundly changed the traditional landscape order in Canton, while the Protestant Dongshan Church interacted more closely with the city.

**Keywords:** Christian churches; Canton; historic urban landscapes; Sino-Western interactions; Protestantism and Catholicism; Christian communities





## 1. Introduction

Canton, a long-standing trading port in the Pearl River Delta, was located on the north bank of the Pearl River that flows into the South China Sea. The geographical advantage of the long-distance river estuary has underpinned Canton's development as a convergence point for Sino-Western interactions. During the Ming and Qing dynasties, as the capital of Guangdong province, Canton was characterized by Confucian hierarchical order and was also an important stronghold for Christian missionaries.

The academic reflection on the history of Christian missionary work in China has highlighted its political role as "necessarily indeterminate" (Dunch 2002, pp. 301–25), and the complexity of missionary activities has been revealed through different research paradigms, in particular, it has summarized the theoretical paths, combining the modernization paradigm with postcolonial theory to explore modern missionary practices in China (Lixin Wang 2002a, pp. 98–109, 2003, pp. 31–37). In view of this, churches, which exist as objects, can serve as appropriate carriers for observing and evaluating the history of missionary work. Churches were not only venues for missionary work but also demonstrated connections with the daily lives of missionaries and Christian believers, as well as urban spaces and landscapes. The location and construction of these churches were related to the political and cultural games between missionaries, Chinese officials, and the public. In different

historical periods, with the rise and fall of Chinese and Western powers and the evolution of their relationship, missionaries adopted different strategies to build churches.

Previous studies have focused on the design and construction of churches, showcasing the details of conflicts or collaborations between missionary societies, Canton officials, and the public during the construction of churches (Masson 2019, pp. 116–61; Coomans 2019, pp. 162–92; Wiest 2004, pp. 231–52), and analyzing the architectural characteristics of churches in Canton (Xue et al. 2009, pp. 48–52; Guo 2018, pp. 70–76; Du 2009; Yin 2021; H. Zhu 2004, pp. 14–16). Other studies have emphasized the interactions among local elite groups, residents, and foreigners in Canton's urban planning and construction, influenced by foreign forces (Conner 2014; Xie and Walker 2022, pp. 685–712; Gu and Hein 2023, pp. 695–708).

Based on the above achievements, this article studies the spatial expansion process of Christian churches as an urban landscape element in Canton after Christianity was introduced into this city, as well as its influences on the city's urban landscape during this process. On the one hand, the church was a core element in traditional cities with Christian beliefs in the West, often occupying a central position in Christian cities or communities. On the other hand, for cities under the Confucian order in China, churches were very different heterogeneous elements, especially those with bell towers, which would have inevitably impacted or even shocked the traditional Chinese urban landscape order that emphasized imperial power. Compared to existing research studies, this article further examines the expansion of churches and their interactions with the city from the perspective of the overall city, providing a new multidisciplinary approach to analyzing and evaluating the history of missionaries and church construction.

The history of Christian missionaries in Canton can be traced back to the late sixteenth century (during the late Ming dynasty) when the main urban layout and appearance of Canton had basically taken shape (Z. Zeng 1991, pp. 344–426). Therefore, it would be much easier to conduct comparative research on the influence of churches on the urban landscape order. From the landscape study approach (Bruns et al. 2015), this article conducts a multi-dimensional comparison and examines diachronic and synchronic perspectives across four aspects.

First, this article derives the main characteristics of the spatial layout and the urban landscapes in Canton within the context of the Confucian order. It presents a framework for understanding the historical expansion of churches and assessing their influence on urban landscapes. This part focuses on the city's functional area distributions and spatial layout, the integral characteristics of urban landscapes, and the construction of the urban space order. In order to facilitate the understanding of the conflict between Christian churches and the urban landscape order of Canton, this article conducts a comparative analysis between Canton and Venice, which is a medieval Christian port city.

Second, this article analyzes the historical process of church expansion in Canton from multiple perspectives (e.g., missionaries, Western tourists, Canton authorities, and local residents) and assesses the influence of churches on urban landscapes. In order to achieve these intentions, this article collects church archives, journals sponsored by missionaries, travel notes and books written by Western tourists, Canton gazetteers, writings by local residents, paintings, and photos at that time.

Third, this article describes the expansion process of the churches in two stages based on the historical background of missionary work. In 1582, Catholic missionary Michele Ruggieri established the Notre Dame Chapel in the Siam Pavilion in the western suburbs, initiating the history of missionary work in Canton (Henri 1936, pp. 190–91). Until the beginning of the Qing dynasty, due to the influence of the "Chinese Rites Controversy" between the Holy See and the Qing Empire, missionary work was interrupted (T. Li 2019, pp. 15–122; Louis 1991, p. 275). In 1732, the Canton authorities demolished the Christian churches and expelled the missionaries (Dai and Shi 1879, p. 396; Louis 1991, p. 275). Based on these cases, the first stage spanned from 1582 to 1732. At this stage, the Qing court had a strong influence as well as the ability to control Western missionaries. The Western mis-

sionaries preached and conducted church construction after obtaining permission from the emperor or local officials. The missionaries also had good relationships with the emperor and local officials and obtained official permission and protection so they could preach (Hu 2003, pp. 6–39). However, after 1732, there were no historical data sources on churches in Canton until 1844, when the Treaty of Wanghia was signed by Sino-the United States, and the Treaty of Whampoa was signed by Sino-France[1] (W. Li 1996, p. 234). These two treaties stipulated that the Americans and the French had the right to rent land and build churches at five trading ports (namely, Canton, Fuzhou, Xiamen, Ningbo, and Shanghai) (T. Wang 1957, pp. 54–62; Yixiong Wu 2000, pp. 114–15; Louis 1991, p. 275). Therefore, for a comparative study, this article ignores the situation from 1733 to 1843 due to the lack of historical literature on churches and classifies the period from 1844 to 1911 as the second stage.

Fourth, this article analyzes the evolution of the layout planning of churches and their influences on urban landscapes by focusing on typical case studies. Regarding the differences in service targets, construction purposes, the division between Catholicism and Protestantism, and the diversities in spatial composition and organization, this article focuses on the following examples: the Thirteen Factories Church and the Shameen Church (both established by British Anglicans), the Roman Catholic Cathedral—also known as the Sacred Heart Cathedral—built by the Paris Foreign Missions Society, and the Dongshan Church (developed by the Southern Baptist Convention.

## 2. The Spatial Layout and the Landscape Characteristics of Canton

In traditional Chinese urban planning, the government controlled the planning and construction of major functional facilities. In the Qing dynasty, cities at provincial, prefectural, and county levels were required to build functional facilities of at least six categories, including administrative, defensive, educational, storage, sacrificial, and relief facilities, forming a spatial pattern and order dominated and controlled by the government (Sun 2023, pp. 405–39; Sun 2021, pp. 20–29). In particular, the cultural and educational buildings related to imperial examinations and social education flourished in urban planning and construction, significantly improving their status in society and urban landscape (Lumin Wang 2002b, pp. 108–21). Moreover, legal regulations had restrictive provisions on the height, width, style, color, and decoration of buildings of different grades and diverse types (Wang et al. 2018, pp. 1180–88). Canton, as a provincial capital, was built under these regulations (Z. Zeng 1991, pp. 344–426).

### 2.1. Urban Functional Zoning and Spatial Layout

Canton was bordered by the Baiyun Mountain to the north and the Pearl River to the south. The terrain was high in the north and low in the south. The whole city was divided into seven parts by the city wall and the Pearl River, and these parts were differentiated by distinct boundaries, respective dominant functions, and hierarchical statuses. From north to south, Canton was divided into the Old City, the New City, Bird-Wing City,[2] the Water City on the Pearl River, and Honan. The west and east sides were the western suburbs and the eastern suburbs, respectively (X. Zeng 2013, pp. 105–9). Canton City maps were found in both Canton gazetteers and the English newspaper The Chinese Repository. Both were published in the 1830s and showed the layout and structure of Canton City (Figure 1). By examining historical maps and related literature, the Old City, as the name suggests, was the earliest planned and constructed urban area among the three cities. It was an area where government buildings, schools, publishing houses, and sacrificial facilities were concentrated. Most of the dignitaries also lived in the southwestern part of the Old City. Moreover, although there were few government buildings in the New City, the high-level Viceroy office of Guangdong-Guangxi and the Superintendent of Customs were located in this district. Banks, guild halls, and industrial markets were nestled in the New City, assuming the economic role of facilitating trade between Canton and the inland cities. Furthermore, there were many wharves in the bird-wing city along the river. They became centers for wholesale markets and served as hubs for vendor transactions in the

city. The western suburbs mainly included the commercial area in Eighteen Ward Street (which first emerged during the Ming dynasty), alongside the Huaiyuan Posthouse, the Thirteen Factories (used for trading with foreigners), and the Shameen concession area built in the 1860s. In the eastern suburbs, business markets trailed along the Pearl River, and there were mainly military training grounds and charity facilities. The eastern suburbs were less developed and relatively smaller in scale than the western suburbs. The Honan Island on the south side of the Pearl River was also developed after the reign of Qianlong (after the 1730s), forming commercial and residential blocks. Haichuang Temple, which was often visited by Westerners during outings, was located on the Honan riverbank. Honan became an emerging industrial area during Daoguang's reign (around the 1820s) and was an important processing site for export commodities of the Thirteen Factories (X. Zeng 2013, pp. 105–9).

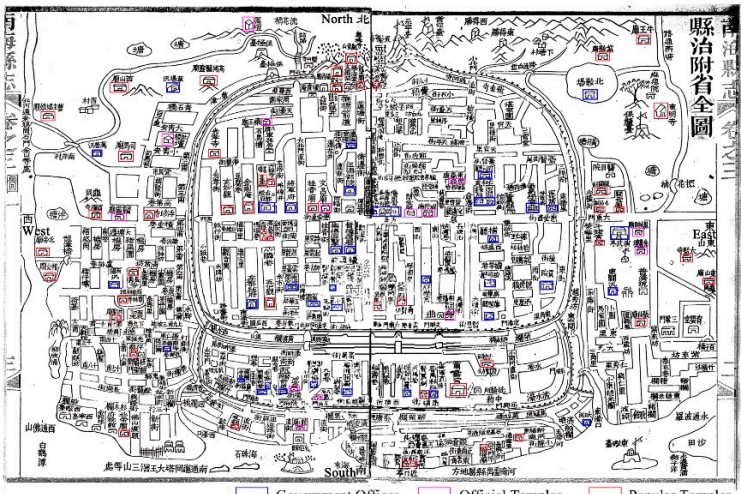 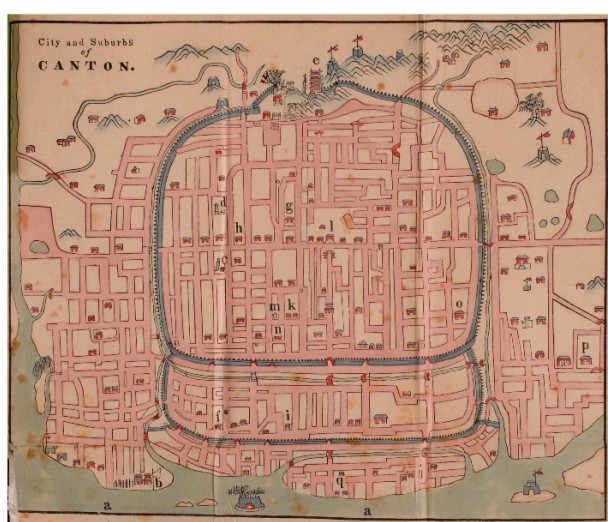

**Figure 1.** Canton Map in the 1830s recorded in *Canton Prefectural Annals and Chinese Repository*.[3] Source: (Lai 2016, p. 193; Bridgman 1833, p. 160).

Furthermore, a large number of boats on the Pearl River in the south of Canton formed a floating city, which Westerners visiting Canton focused on, even though they were marginal characters seldom mentioned in local chronicles. In 1751, Peter Osbeck wrote in a letter that the boats approaching the city of Canton were close to each other, resembling crisscrossed streets (Osbeck 1771, p. 227). When Charles Toogood Downing arrived in Canton in 1836, he was shocked by the endless variety of boats on the river. Downing observed countless boats of different sizes, shapes, and colors plying back and forth on the river, and the boats were inhabited by craftsmen and traders engaged in various professions, constituting a wonderful water world that was completely different from anywhere else on earth (Downing 1838, pp. 103, 221–24). However, contrary to the desirable descriptions by Westerners, water dwellers were derogatorily called the Dan people (疍民, those who lived in egg-like boats), a socially underground and marginalized group of people. The Dan people had been making a living on boats for generations. They had no land-registered residence and were not permitted to take imperial examinations to achieve class promotion. People on the land even refused to marry them (R. L. Wang 2020, pp. 58–59). In the eyes of Westerners, the glamorous floating boats were only marginal, and they were far less significant than mountains, rivers, city walls, and monuments. Therefore, it was almost impossible to see such vivid and detailed descriptions of these boats and water dwellers as Westerners did in local gazetteers (Ni 2007, p. 90).

There were two key differences in the functional zoning and layout of Canton City compared to the medieval Christian city of Venice. First, Canton was dominated by the Old City and the New City, where the government was concentrated. The other five functional

areas were subordinate and the public facilities in each area were scattered without a clear functional aggregation center. Differently, Venice was composed of six functionally homogeneous urban precincts (or parishes), and each precinct was centered on a church. The churches were usually surrounded by squares, fountains, guild halls, and schools, which made the political status and the landscape of the parish center more prominent (Mumford 1966, pp. 307–8, 368–70). Second, the government buildings of Canton remained far away from the Pearl River coastline, while the city center of Venice, St. Mark's Basilica, and Doge's Palace were close to the coastline and faced the bay (Benevolo 2000, pp. 602–5) (Figure 2).

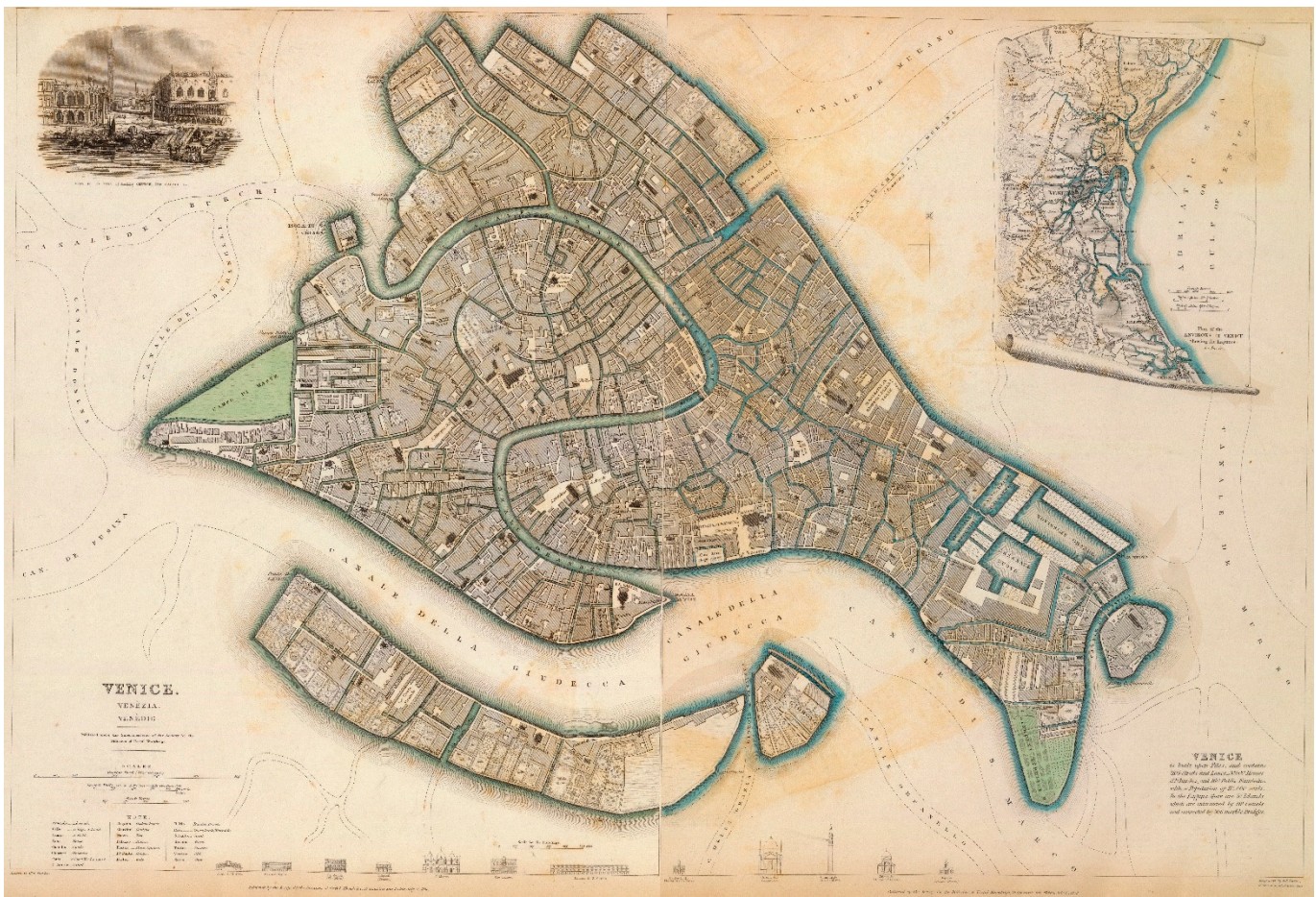

**Figure 2.** Map of Venice in 1838. Source: (Society for the Diffusion of Useful Knowledge 1844, pp. 209–10) (drawn by W.B. Clarke).

### 2.2. Landscape Features with Horizontal Extension

Significantly different from the high-rise development of buildings in medieval cities (Benevolo 2000, p. 354), the low houses in Canton were spread out on the ground level. When the Portuguese first arrived in Canton in the sixteenth century, they found that the ordinary residential houses and the noble residences in Canton were all one-story (João de Barros 1563, p. 222). In the 1750s, the houses that the Jesuits saw were still single-story, and they even thought that this was a popular style in China (Ripa 1855, p. 34). In the 1790s, Anderson observed that there were two-story buildings in Canton's foreign trade port, but buildings in Canton were still generally short (Anderson 1795, p. 38; Winterbotham 1795, p. 104). In the 1850s, George Wingrove Cooke, the correspondent of *The Times* stationed in China, commented that—except for pagodas, temples, and state bureaus in Canton—there was almost no other building higher than the lowest house on Holywell Street. The

residential houses were generally about 15 feet (about 4.6 m) high, and the highest two-story shops were no more than 25 feet (about 7.6 m) (Figure 3) (Cooke 1858, p. 75).

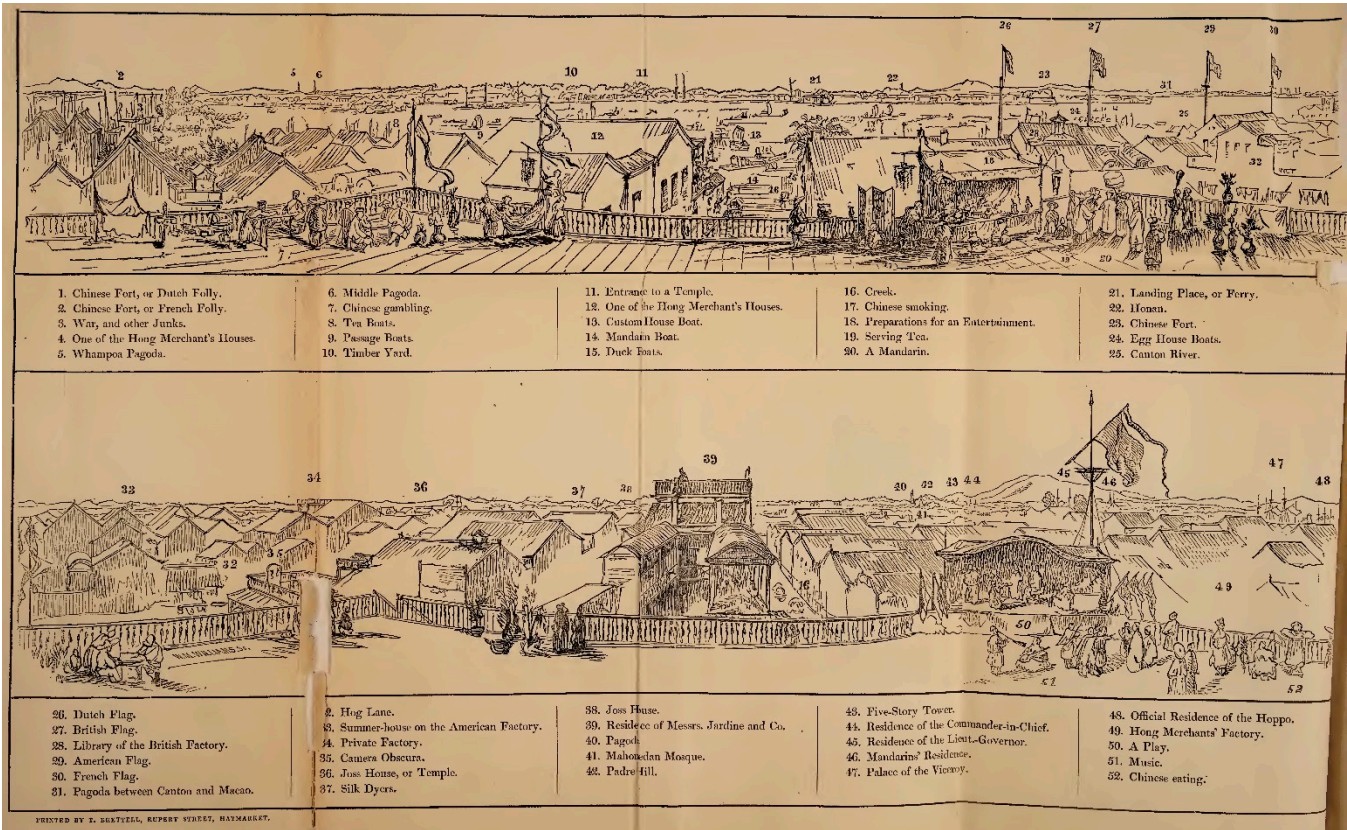

**Figure 3.** Description of a View of Canton (1838).[4] Source: (Burford 1838).

Compared with Venice, judging from the photos of Venice in the 1870s, St. Mark's Basilica, as the city center, and adjacent to the Port Square, obviously supported the supremacy of Christianity and the Patriarch of Venice. The towering St. Mark's Campanile echoed the bell towers of other parishes, drawing visitors' eyes skyward. Judging from the photos of Canton in the 1860s, the city of Canton was spread out horizontally and lacked visual focus. The lack of variation in the city skyline directed views to move horizontally. The overall visual status of most buildings was balanced. Under official control, pagodas and minarets, because of their lower political status, were kept at a certain distance from the streets where government buildings were concentrated (Lumin Wang 2002b, p. 56) (Figure 4).

Three pagodas along the Pearl River and to the east of the city, namely Chigang Pagoda, Pazhou Pagoda (or Sea Turtle Tower), and Lotus Pagoda, further extended the horizontal landscape image of Canton City. As a product of *fengshui* planning by officials, these pagodas, together with the Baiyun Mountain, the Pearl River, the Smooth Pagoda, and the Flowery Pagoda in the Old City, formed a *fengshui* landmark system from Canton to Boca Tigris (Humen), carrying the aspirations of the elites to strengthen the momentum and literary fortune of the provincial capital (Qu 1991, pp. 445–46; Qiu 1993, p. 70). According to the accounts of Westerners, a person standing on the Zhenhai Pagoda could still see the Chigang Pagoda and Pazhou Pagoda, although they were far apart from each other. In turn, Westerners who went up the Pearl River were often attracted by the pagodas, and combined, these pagodas became a landmark, signaling their approach to Canton (Hunter 1885, pp. 195–96; Noble 1762, p. 284).

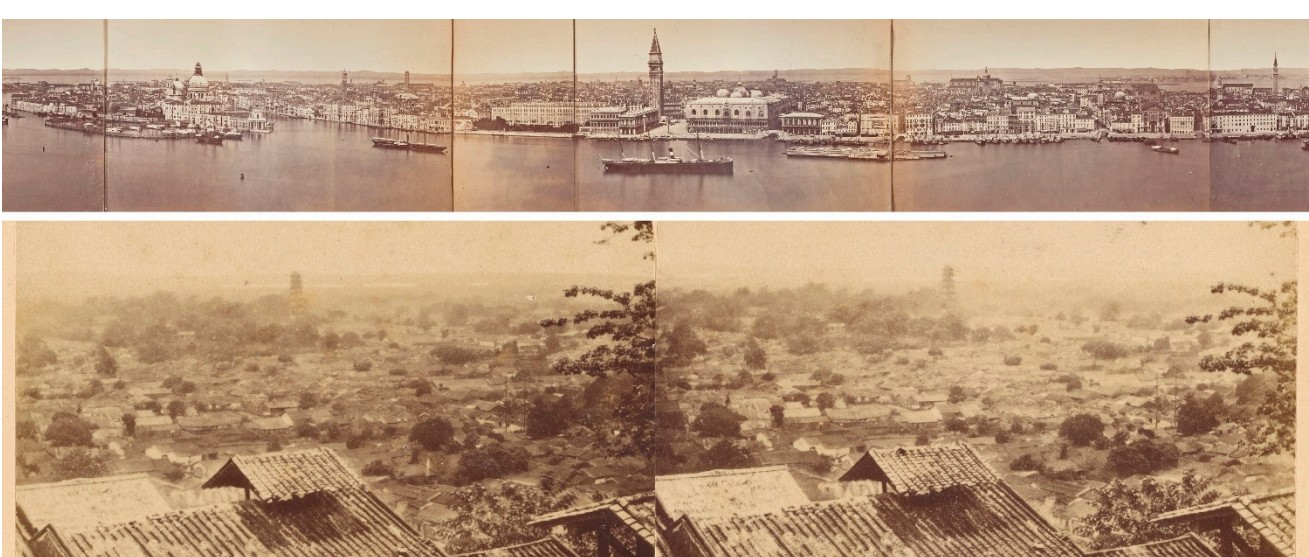

**Figure 4.** Urban landscape comparison between Venice and Canton in the middle and late 19th century. Source: The photo of Venice: https://commons.wikimedia.org/wiki/File:Panorama_of_Venice_1870s.jpg (accessed on 30 July 2024).[5] The photo of Canton: https://www.getty.edu/art/collection/object/107F0H (accessed on 30 July 2024).

### 2.3. The Construction of Canton's Space Order

Canton lacked centrality in its overall urban form, and the spatial order presented a balanced and coordinated state horizontally. Medieval cities emphasized the core role of the city center by means of its winding streets, and the Baroque city radiated straight avenues and regular block units from the central square (Mumford 1966, pp. 349, 441–47; Benevolo 2000, p. 352). Comparatively, Canton possessed the characteristics of both; the main streets leading to the city gate were relatively long and straight while the other roads were mostly short and curved (Ni 2007, pp. 49–50). The government buildings, temples, schools, and other public buildings in Canton City were arranged in an east–west or north–south sequence along the streets; in particular, the main government buildings were mostly situated on the north side of the Avenue of Benevolence and Love, which connected the east and west gates of the Old City. Under the rules of the building grading system, due to the restrictions on the volume, style, and color of these buildings, it was difficult to receive attention from a distant perspective. Although individual buildings may have appeared to be mediocre to some Westerners, as British diplomatic envoy Earl George Macartney commented, "upon the whole, it often produces a most pleasing effect; as we sometimes see a person, without a single good feature in his face, have, nevertheless, a very agreeable countenance" (Bridgman 1833, pp. 194–95).

The prominent positions of important buildings in the urban landscape of Canton were achieved via layout planning of the city and block. Taking advantage of several consecutive honorary portals, Canton City reflected the grandeur of the important streets where the government buildings were situated (Wang and Xiao 2016, pp. 72–77), forming a city axis in front of the key government buildings. At the entrance of important government buildings, special structures, including memorial archways, screen walls, flagpoles, and stone lions, were used to emphasize the authority of these buildings, which seemed more prominent than other general buildings. The north–south vertical arrangements of official buildings also shaped their own axes (Figures 5 and 6), forming the spatial order of the city (from urban and architectural perspectives). In comparison, with certain exceptions, there were no formal axes in front of important medieval buildings (Mumford 1966, p. 351).

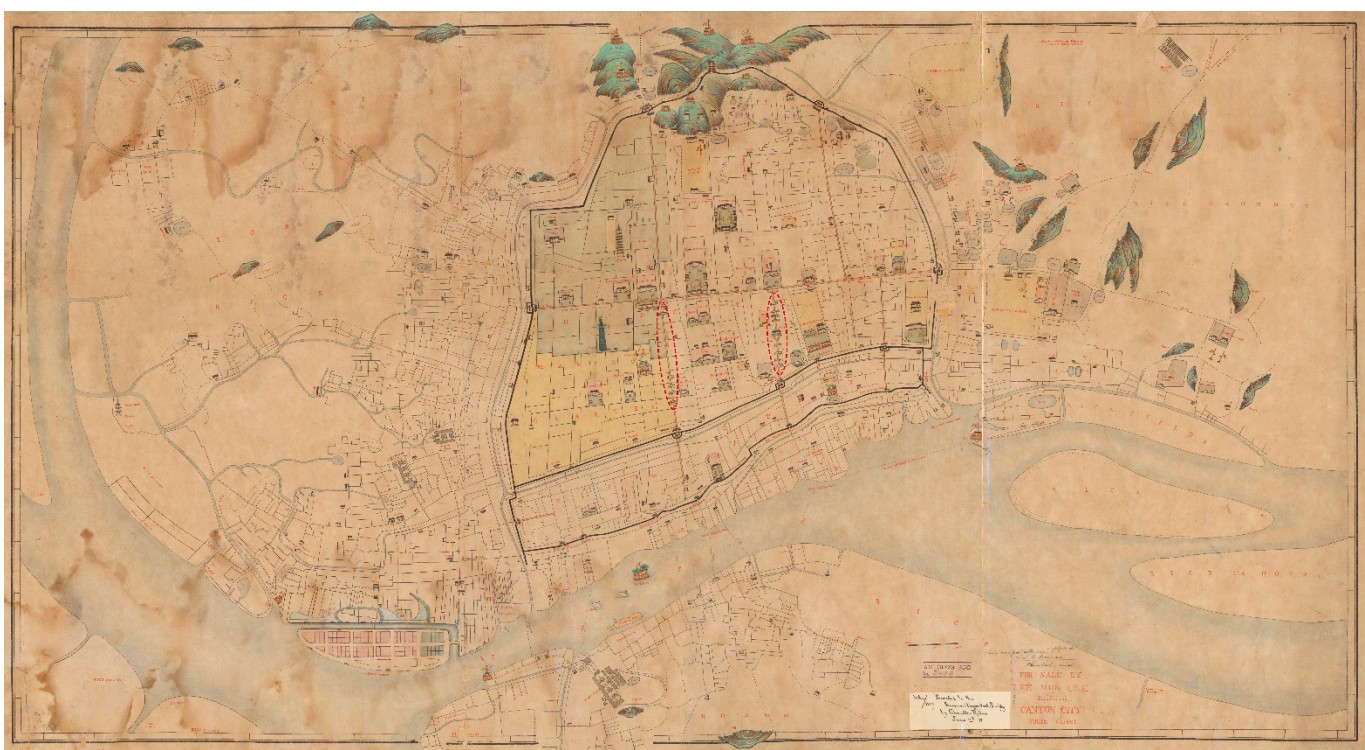

**Figure 5.** Canton map in 1862. Source: https://collections.lib.uwm.edu/digital/collection/agdm/id/7502/ (accessed on 30 July 2024).

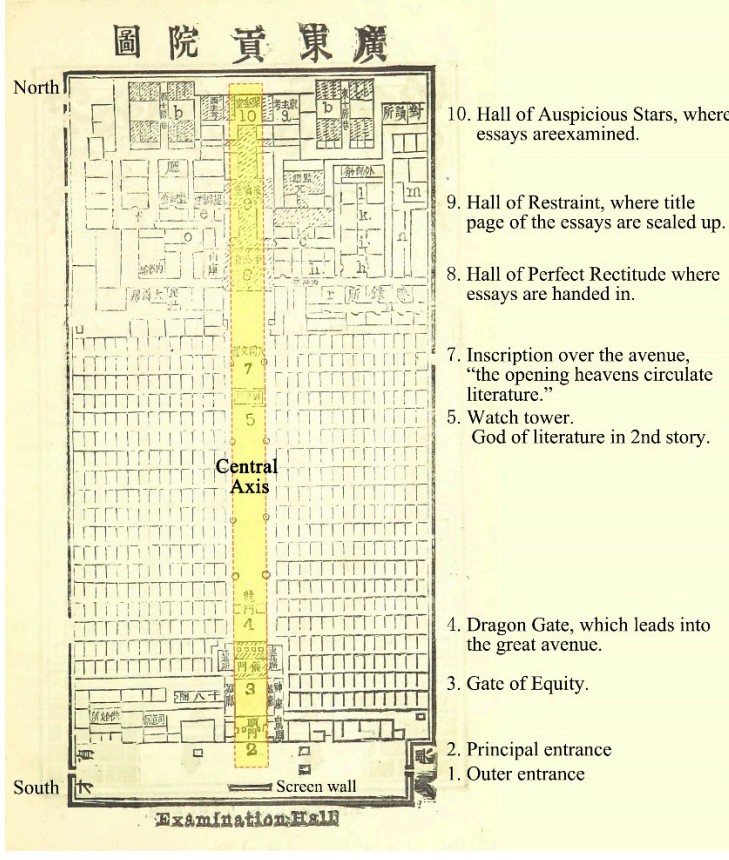

**Figure 6.** The central axis of the Examination Hall of Guangdong Province. Source: (Kerr 1880, p. 16).

Overall, there were significant differences in the functional spatial layout logic and architectural hierarchy rules between Canton and the Christian cities. These differences fundamentally determined that—after the churches with bell towers had entered Canton—there was a conflicting relationship between these churches and the original urban landscape order.

### 3. The First Appearance and the Distribution of Churches in Canton under Dynastic Power (1582–1732)

In 1582, Jesuit missionary Michele Ruggieri established the Notre Dame Chapel in the Siam Pavilion in the western suburbs of Canton (Henri 1936, pp. 190–91). Later, thanks to the tolerant policies toward missionary work in the Southern Ming regime and during the reigns of Shunzhi and Kangxi of the early Qing, the scope of Catholic missionary work in Canton expanded significantly (Hu 2003, pp. 17–39). At the end of the 1640s (when Canton was under the rule of the Southern Ming regime), Francesco Sambiasi was allowed to build a church and a residence within the walled city of Canton, achieving what previous missionaries had sought but failed to accomplish—breaking through the restrictions of the city wall and entering Canton City (Pfister 1932, pp. 136–42; Xiao 1931, pp. 229–30).

Father Adrien Launay, a member of the Paris Foreign Missions Society, recorded seven churches in Canton in 1704. They were Sin-keou-lei (上九里), Kan-wouan-lei (锦云里), Yoc-yu-tong (育婴堂), Tsoc-tong-yamoun (左堂衙门), Tsim-lo-kong-koun (暹罗贡馆), Tai-toc-hang (提督行台), Tai-fat-se (大佛寺) (Launay 1917, pp. 1–3; Coomans 2019, pp. 62–192). As a whole, these seven churches were located in the southern half of the city near the Pearl River, which coincided with the path from the Pearl River to Canton. The first two churches were named after the street names and were situated in the western suburbs of Canton where the traditional business district was located. Yoc-yu-tong Church took its name from *the Foundling Hospital*, and judging from the time, this church should have been built in 1697. It was located in the Seventh Ward outside of the west gate (Bridgman 1833, p. 263; Li et al. 2003, p. 154; Ruan and Wu 1836, pp. 353–54).[6] Tsoc-tong-yamoun Church obtained its name from the Viceroy's Office of Nanhai County. It was originally located on the east side of the Viceroy's Yamum in Nanhai County in Zaoheng-Fang (早亨坊) inside the Guide Gate of the Old City. In 1731, the county magistrate's office was relocated to Yangrenli Alley in the western suburbs outside Taiping Gate of the New City (Pan and Deng 1869, p. 16). Tsim-lo-kong-koun Church was named after Siam Posthouse or Huaiyuan Posthouse. Huaiyuan Posthouse was also situated within the western suburbs and was a posthouse for receiving foreign envoys and undertaking foreign trade. The Thirteen Factories—famous in later generations—were just located on the south side of Huaiyuan Posthouse (Wang and Xiao 2022, pp. 80–93). Tai-toc-hang Church was located near the Admiral's Office, which was originally located in Yuxian-Fang (育贤坊) on the east side of the South Gate of the Old City. In 1799 (the fourth year of Jiaqing's reign), it was relocated to Tianma Lane on the east of Yongqing Gate in the New City (Li et al. 2003, p. 150). Tai-fat-se Church was located near the Great Buddha Temple on Siqian Street (which was the street before the temple) on the west side of the South Gate of the Old City (Figure 7). Five of the seven churches were named after government buildings, relief facilities, or Buddhist temples. Although it cannot be directly proven that the churches utilized the houses of these offices, facilities, or temples, it can at least be indicated that the churches were adjacent to them.

Three of the seven churches mentioned above in 1704 were located in the Old City, adjacent to government buildings or Buddhist temples. On the one hand, this indicated that both the authority and the Buddhists showed a sense of tolerance toward the Christian Missionary Society. On the other hand, the streets where government buildings and Buddhist temples were located tended to observe the crowds, which increased the opportunities for missionary work. At the same time, they kept a distance from the Avenue of Benevolence and Love, where many high-level government buildings were concentrated; this implied an attitude of caution. In addition, the four churches located in the western suburbs be-

gan near the Siam Pavilion and were closely tied to the path of commerce. In particular, the Yoc-yu-tong Church showed that the missionaries paid attention to urban charity and the lower-class residents. In November 1722, Jesuit missionary Father Gaubil quoted the words of Father du Baudory in a letter to the Archbishop of Toulouse, Mr. de Nemond. The letter was about the priest of his church, who baptized critically ill and abandoned babies (Du Halde 2001b, pp. 281–85). These situations showed that—after more than half a century of operation—the missionaries had a better understanding of the functional facilities, spatial layout, and power space of Canton, and these churches were integrated into the public space of the city in an active and cautious way.

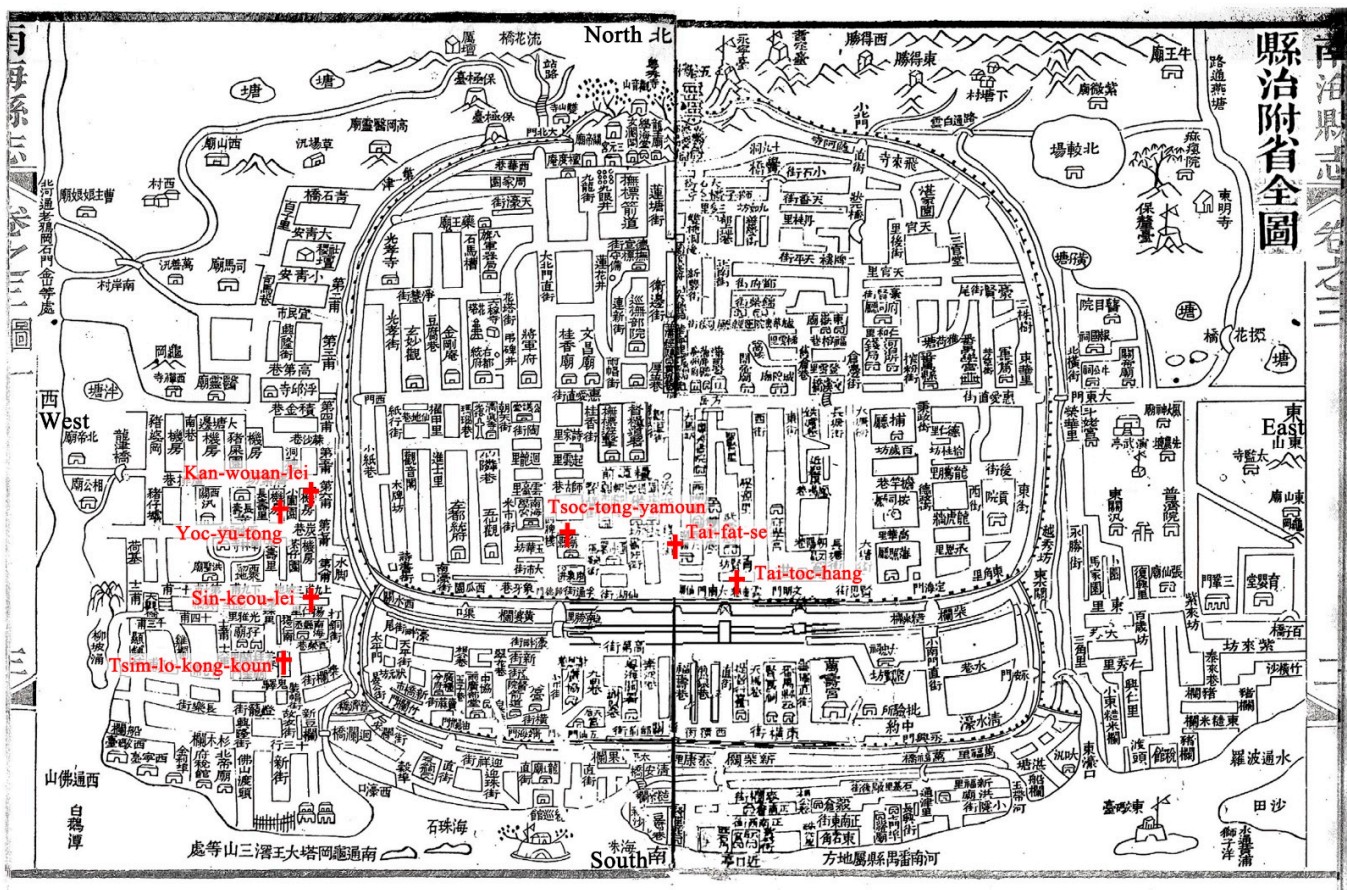

**Figure 7.** Distribution of the seven churches in Canton in 1704.[7] Source of the original picture: (Lai 2016, p. 193).

The architectural style of the churches was also a matter of note. When Alexander Hamilton came to Canton in 1703, he commented: "The Italian Church makes a handsome Figure, but the French Chapel is but mean on the outside" (Hamilton 1995, pp. 5–8). Hamilton's statement showed that the Italian church should have features of a Western church. The Italian church provoked the dissatisfaction of non-Christians, who considered such a church an insult to their houses and pagodas; they complained to the governor, but the governor vetoed the objection (Du Halde 2001a, p. 194). The governor's neglect of residents' resistance to the Italian church indicated that—under the conditions of strict architectural hierarchy and the government's absolute control over Western missionaries at the time— even the Italian church that Westerners considered magnificent was within the scope of official acceptance.

**4. The Planning and Spatial Expansion of Churches and Their Impact on Urban Landscapes under the Weak Situation of the Qing Dynasty, 1844–1911**

After the Opium War, as the Qing court weakened, many countries—such as the United Kingdom of Britain, France, and the United States of America—competed to gain entry into Canton. With support from Western powers, Catholicism and Protestantism spread their faith and built churches in Canton (W. Li 1996, pp. 176–90, 234–43). It would be more appropriate to reveal different strategies adopted by Catholicism and Protestantism by analyzing the expansion, planning, and impacts of the churches on urban landscapes in chronological order.

*4.1. The Expansion of Churches from the Pearl River Bank to the New City, the Old City, and the Suburbs*

From 1844 to 1857, due to resistance from the residents in Canton, foreigners were still unable to settle down and preach in the walled city (Bridgman 1849a, pp. 216–22; Liu 1997, p. 13; Wiest 2004, p. 232). Church construction activities were carried out along the river in the western suburbs and the southern suburbs. This area was the place that Westerners had the most contact with and were the most familiar with, and Westerners often visited it in the sixteenth and seventeenth centuries. In 1845, Liang Afah, a Chinese follower of the London Missionary Society, presided over the construction of a three-story chapel, called the *temple of the true god*. This chapel, facing the main street, was located on the riverside in the western suburbs and was close to the city gate. It was about one mile away from the foreign factories, and a large number of visitors were observed by this novelty building (Bridgman 1846, pp. 134–35). In 1845, Issachar Jacox Roberts, a missionary of the Southern Baptist Convention, rented land in Dongshijiao (东石角), Tianzi Pier of the Southern Suburbs, and set up a lecture hall. Jehu Lewis Shuck, who had a competitive relationship with Issachar Jacox Roberts, also built a church on Pwanting Street (联兴街) at the west end of the Thirteenth Factories Street in the same year. In 1846, Issachar Jacox Roberts purchased a Zidong-ting (a kind of local boat) and preached on the Pearl River. However, the missionary boat was shot and sunk a year later (Liu 1997, pp. 8–9). Of particular note was the Thirteen Factories Church located in the western suburbs. It was constructed by the British Consul in 1847 and completed the following year. The Thirteen Factories Church was built in the region that was managed by the UK, and the Chinese were not allowed to enter or exit freely. Later, less than 10 years after its construction, this church was destroyed along with the Thirteen Factories during the Sino-British War in 1856–1857 (Conner 2014, pp. 198–201). M. Guillemin, a member of the Paris Foreign Missionary Society, who reached Canton in October 1848, built a simple chapel in his house on the Thirteenth Factories Street, which was often visited by European businessmen and Chinese followers (Launay 1917, p. 3; Chen 2008, pp. 31–32). In the 1850s, there was also a floating Bethel open to the public in Whampoa Port, positioned on the outer port of Canton City (Bridgman 1850, p. 168).

In the last days of 1857, the British army bombarded the city of Canton. After occupying the city and suburbs, they planned to rebuild the factories. Harry Smith Parkes, who was in charge of the construction, was unwilling to build a commercial district on the original site as other countries might have claimed the land. So, construction of the Shameen concession area (located about 90 m upstream from the former site of the Thirteen Factories) began under Harry Smith Parkes. This area was called an island as it was surrounded by a river. Four-fifths of the island in the west was occupied by the British Concession and one-fifth in the east was occupied by the French Concession. The Anglican Christ Shameen Church, built around 1865, was one of the earliest buildings in the Shameen Concession (Conner 2014, pp. 231–35). Compared to the previous Thirteen Factories Church, the importance of the Shameen Church had significantly increased as it was built in a formally signed concession area.

A map before 1878 in German marked the distribution of three Protestant churches and two Protestant chapels on the river banks of the southern suburbs and the western

suburbs. Among them, the English Church (the Shameen Church) was in Shameen. The Chinese Church of the London Missionary Society was located by the river on the north side of Shameem and adjacent to the Wesleyan Methodist Missionary Society chapel. The Rhenish Missionary Society chapel was located on the west side of the south suburbs, and on the east side of the south suburbs, there was another Wesleyan Methodist Missionary Society church (Coomans 2019, p. 186). On this map, of particular note, was the Roman Catholic Cathedral, which broke through the limitations of the city walls and was located at the original site of the Governor General of Guangdong and Guangxi in the Old City. The former Governor General's Office was destroyed by gunfire in 1857. The French disregarded the opposition of the Canton authorities and surrounded this land with troops. Under the protection of French soldiers and cannons, the Roman Catholic Cathedral began construction in 1863; the construction lasted for 25 years until its completion in 1888 (W. Li 1996, p. 191). This land was sandwiched between the north and the south by the city walls, and the surrounding development space was limited, making it a less-than-ideal site for a missionary center. A reasonable explanation was that its declaration value was the main factor of this choice (Figure 8).

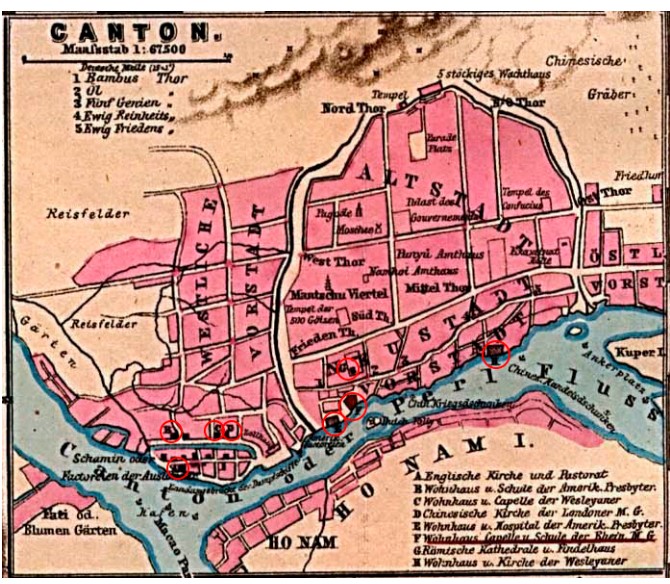

**Figure 8.** Distribution of churches in Canton (1863–1878).[8] Source: http://www.lib.utexas.edu/maps/historical/kwang_tung_1878.jpg (accessed on 30 July 2024).

After the Roman Catholic Cathedral had broken through the limitations of the city walls, the Protestant churches quickly expanded to the New City, the Old City, and the suburbs. A map published in 1880 identified fifteen Protestant churches (or chapels) and gospel halls, which were relatively scattered overall. Among them, the three churches in the Old City were close to the two avenues, namely the east–west avenue of Benevolence and Love, and the north–south street of the Four Archways. Another two churches in the New City were also respectively situated on the east–west main streets, namely High Street in the east and Green New Street in the west.[9] The site selections of these five churches could better integrate with the living contexts of residents while also maintaining certain distances from the government and temples. This was different from the situation in 1704, when three churches in the city were located close to government buildings or temples. They were likely deliberately doing this to avoid arousing strong resistance among the officials and the citizens. In Eighteen Ward of the western suburbs, close to the inland and even further north, another five churches were built, and all settled on the main streets. A Protestant chapel was also built in Honan, which could have been related to the development of Honan and the increasing arrival of Westerners (Figure 9).

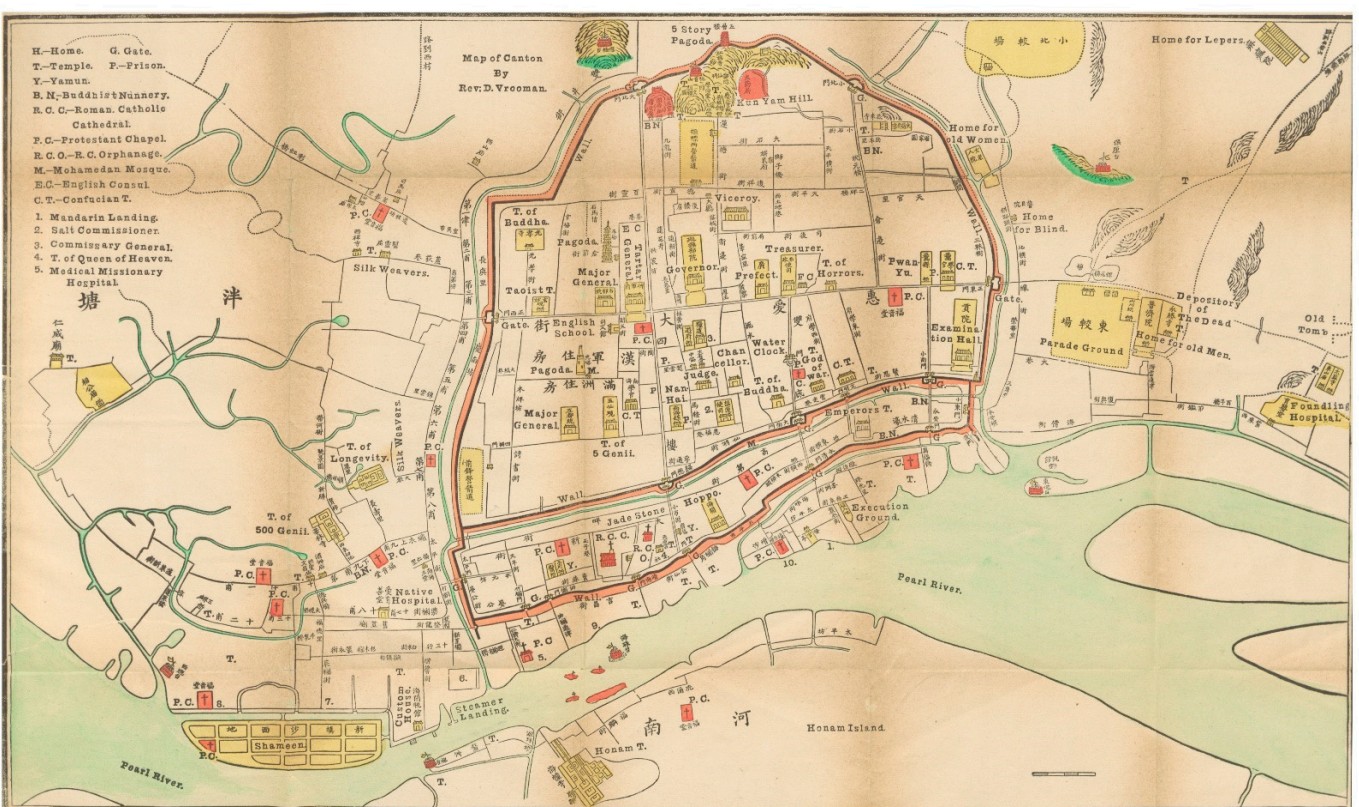

**Figure 9.** Map of Canton with churches and government offices marked (about 1880). Source: (Kerr 1880, p. 25).

After the 1890s, there was another church in the French Concession of Shameen, built by the Foreign Missionary Society of Paris from 1890 to 1893, known as Notre Dame de Lourdes. Therefore, the practice of preaching in the Pearl River also continued. Rev H.J. Von Qualen, a member of the Swedish Evangelical Free Church of the USA, led the construction of a gospel boat called the *Morning Star* in 1896 (Liu 1997, pp. 8–9, 19). The eastern suburbs were not ignored by the missionaries. The American Southern Baptist Convention chose the Dongshankou (东山口) area as a base for missionary activities in 1906 and built the Dongshan Christian Church in 1909 (Z. Zhu 2019, pp. 202–3).

*4.2. The Layout Planning Progress of the Churches*

According to the church floor plans of different periods, as the Western influences continued to deepen and the scope of missionary work continued to expand, the layouts of the churches changed from single buildings to parish-centered, function-mixed building complexes. The Anglican Thirteen Factories Church, built in 1847, stood in isolation on an empty land surrounded by rivers. On this land, there were no other buildings except for the church. The north side of this land was connected to a garden by bridges. On the other side of the garden stood the Thirteen Factories. The south side was close to the riverside and connected to the Pearl River by piers. The bell tower of the church faced the Pearl River in the south, which was completely different from the traditional orientation of Western churches, where the entrance was on the west and the altar lay on the east (Taylor 2003, p. 23). Moreover, since the entrance of the church was on the south side, when attending services, westerners who lived in the factories on the north side of the church needed to go through the garden to the south side of the church to enter, which added extra inconvenience (Figure 10). It is easy to see that such a design underscored the evolving status of Westerners; the church was a symbol of the colony, and political significance was just as important as religious significance (Conner 2014, pp. 184–85).

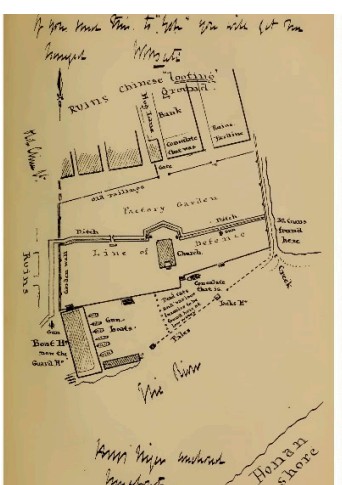 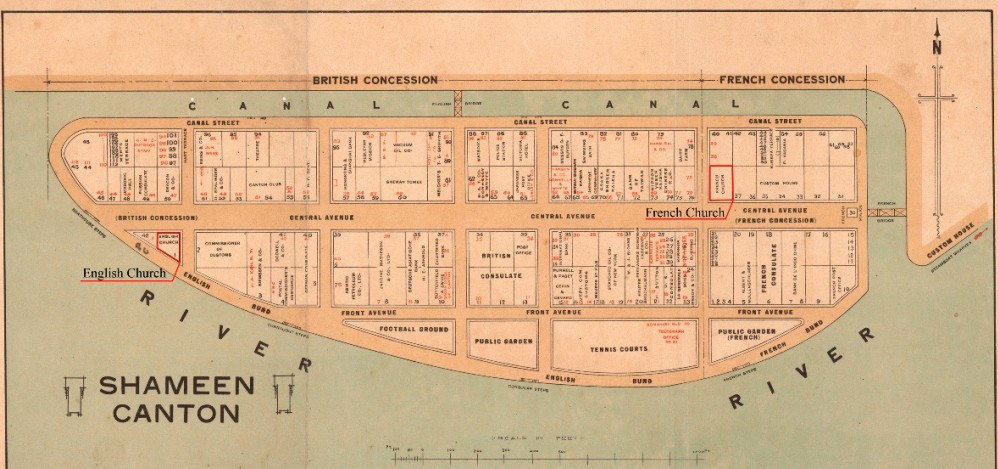

**Figure 10.** Layout of the Anglican Thirteen Factories Church and the Anglican Shameen Church. Source of the left part: (Lane-Poole 1894, p. 254). Source of the right part: https://nla.gov.au/nla.obj-229820164/view (accessed on 30 July 2024).

The Anglican Shameen Church adopted the same layout as the Thirteen Factories Church (Figure 10). Like the Thirteen Factories Church, the bell tower of the Shameen Church faced the Pearl River. The difference was that the Thirteen Factories Church was situated in the middle of the site, while the Shameen Church was located on the southwest corner of the island. Residents of the island, especially those on the east end of the island, had to walk a little longer to reach the church, which became even more inconvenient. However, this direction was not only opposite to the main channel of the Pearl River but also faced the waterway on the west side of Honan. In other words, the Shameen Church, positioned opposite the intersection position of rivers, had a more prominent propaganda role. However, the Thirteen Factories Church and the Shameen Church were located in relatively independent areas and served Christian believers in Western neighborhoods. From this point, the two churches were introverted and detached from the urban context of Canton.

The Roman Catholic Cathedral underwent a fundamental transformation. Its construction opened up a new model in Canton—planning and constructing a parish-centered church complex with mixed functions on a block scale, similar to the parish center of a medieval city. An ambitious church-building plan was first spelled out in a letter in 1852 by Father M. Guillemin,[10] from the chapter of the Paris Foreign Missions Society (Wiest 2004, p. 250).[11] According to the plan by M. Guillemin, there was also a Jesuit College, an orphanage, and a missionary's residence around the cathedral. Across the street from the cathedral, there was one school for males and another for females (Coomans 2019, p. 177). On the eve of the completion of the Cathedral, Father M. Guillemin passed away. His successor, the new bishop Augustin Chausse, built the bishop's mansion and a theological seminary to provide education for Chinese godfathers, according to the plan by M. Guillemin (Masson 2019, p. 157). After that, a male school, a female school, an orphanage, and a craft workshop were newly built. A square facing the entrance of the Cathedral was also built across the street from the cathedral (Figure 11). Although the neighborhood where the cathedral was located was surrounded by a wall, most believers lived near the cathedral (Wiest 2004, p. 243) and established social and economic ties with Canton City through male and female schools, orphanages, and craft workshops. In particular, the male school provided new hope for improving the status of women with low education levels (Smith 1847, pp. 23–25, 120), although there was strict gender segregation in the community. This development essentially formed a Catholic community.

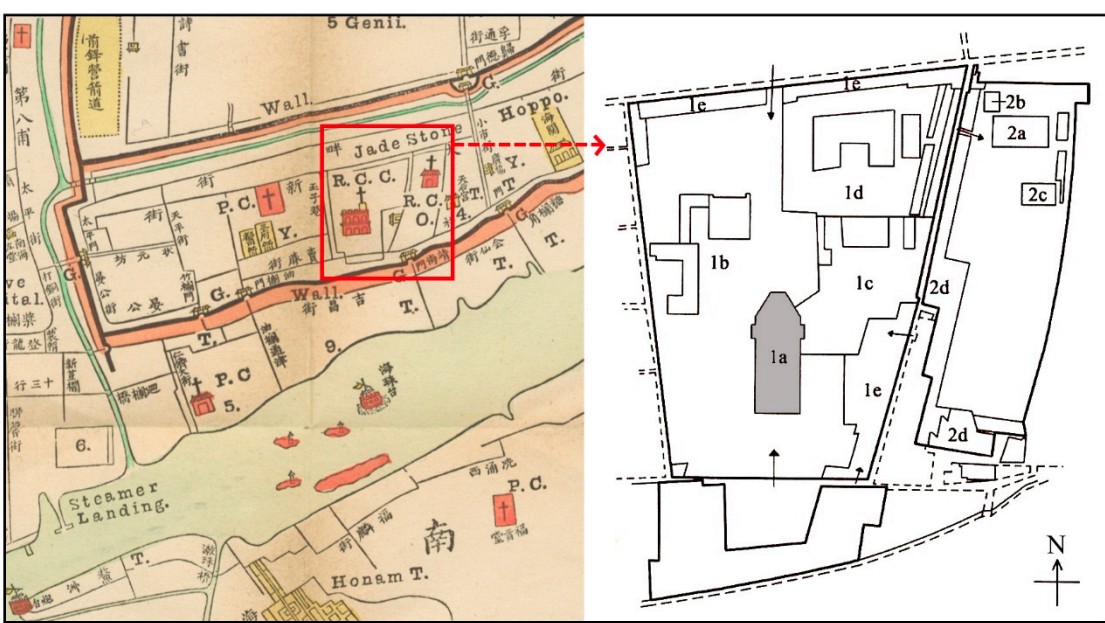

**Figure 11.** Location of the Roman Catholic Cathedral and the architectural complex plan.[12] Source of the left part: (Kerr 1880, p. 25); Source of the right part: (Coomans 2019, p. 190).

Of course, it is worth noting that in the 1840s, Protestant missions had already opened hospitals and schools to facilitate missionary work. At that time, no public missionary activities could be held in the local community, and churches converted from local houses were often destroyed by local residents (Smith 1847, pp. 23–25, 100–1). The church distribution map before 1878 (Figure 8) also recorded a hospital (the symbol 'E') and a school (the symbol 'B') from the American Presbyterians. In the church distribution map of 1880 (Figure 9), these two locations were marked with the symbol 'P.C.' ("Protestant Chapel"), while a Methodist Church of Great Britain chapel and the London Missionary Society Chinese Church on the east side of the Presbyterian School had disappeared. Compared with the Roman Catholic Cathedral's building complex, these scattered, small-scale chapels, hospitals, and schools were not enough to affect the urban layout.

The Southern Baptist Convention, in conjunction with the Chinese branch, began to purchase land extensively in the Dongshan (East Mountain) area in 1906. They successively built the Christ Dongshan Church, churches established by the Chinese themselves, theological seminaries, Pooi To School (male), Pooi To School (female), Pui Ching Academy, Pei Xian Bible School (female), the Muguang Blind School, the Southern Baptist Convention publishing house, the foundling hospital, a nursing home, hospitals, and residences for Chinese and foreign missionaries (Editorial Committee of Gazetteer of Dongshan District 1999, p. 78; N. Wu 2007, pp. 78–81; Z. Zhu 2019, p. 203). The main entrance of the Dongshan Church was set up in the east, while Christians faced the west during worship; the bell tower was in the southeast corner. This design not only adhered to the conventional orientation but also presented a complete and towering image facing the Pearl River, especially for the route from the Huangpu Port in the east to Canton City. It is important to emphasize that, compared to the Roman Catholic Cathedral community, the Dongshan Church community added a hospital and publishing house as public facilities. The layout planning of the Dongshan Church was more open as the church and its ancillary facilities were spread out along the four streets, rather than being limited to one or two blocks (Figure 12).

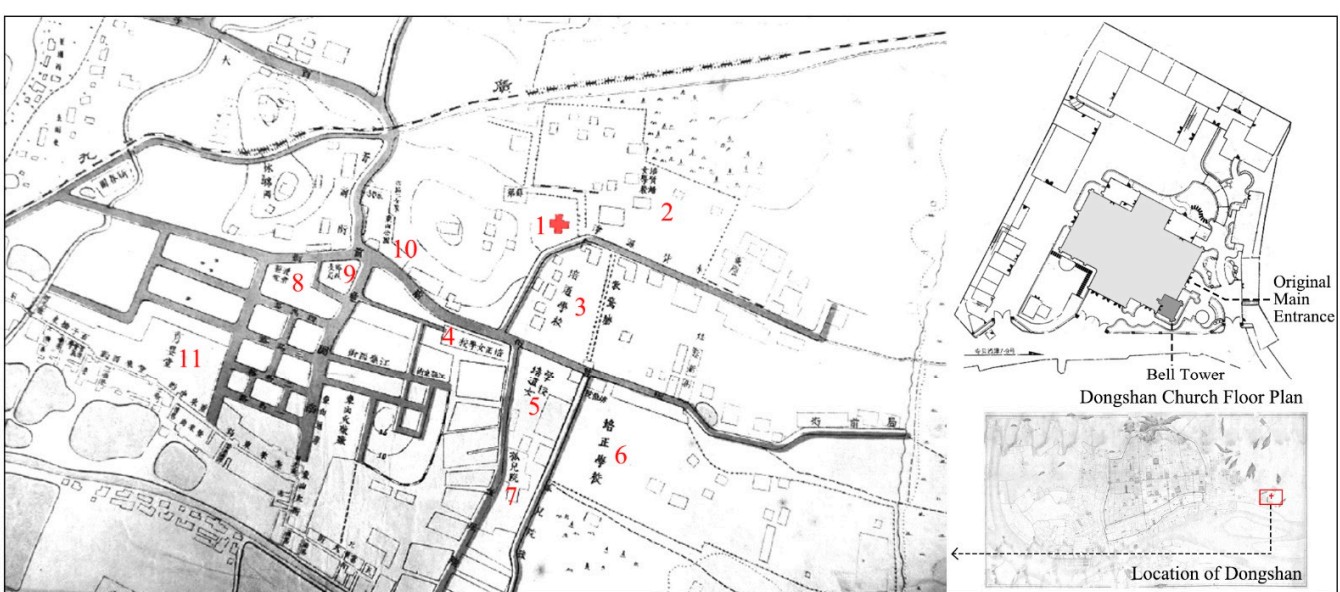

1. Christ Dongshan Church
2. Pei Xian Bible School (Female)
3. Pooi To school (Male)
4. Pui Ching School (Female)
5. Pooi To school (Female)
6. Pui Ching School (Male)
7. Orphanage
8. Baptist Hospital
9. Post Office
10. Dongshan Park
11. Foundling Hospital

**Figure 12.** Plan of Dongshan Community and the floor plan of the Dongshan Church in 1923. Source: (Cao 2019, p. 112); The floor plan of the Dongshan Church is copied from the notice board at the main entrance of the church; The location of Dongshan refers to Figure 6.

The community-based planning and layout of the Roman Catholic Cathedral and the Protestant Dongshan Church presented different ways of life and spatial organizations to the general population in Canton, establishing a foundation for competing with the authorities for spatial power. The churches ventured into constructing educational and relief facilities that belonged within the scope of the government's power and responsibilities and added modern hospitals, schools, and publishing houses. The churches could also contend against official sacrificial facilities and folk temples. As a result, a new parish-style spatial order organization centered around the church was created, which had a strong influence on Canton's urban planning and development. The history of the Dongshan area in the early 20th century proves that the church community in Dongshan had driven the land development and urban construction of the Dongshan area, with the open layout of the Dongshan Church community having a wider, deeper, and longer-lasting impact on the city (Editorial Committee of Gazetteer of Dongshan District 1999, p. 78).

*4.3. The Influences of Churches on Urban Landscapes*

Corresponding to the evolution of the churches' layout planning, the influences of these churches on the urban landscape had expanded from partial areas to the overall spatial order of the city. The church shown in the earliest photos and paintings is the Anglican Thirteen Factories Church. This church was equipped with semi-circular windows and two rows of side aisles. It was in Italian style, about 13 m wide, and about 25 m long; the bell tower was about 20 m high (Bridgman 1849b, p. 112). The Thirteen Factories Church was a fine complement to the European-style houses on its north side, and together with the garden, they presented a unique Western style. Although it was placed in front of the Thirteen Factories and its height and style were also eye-catching, looking at the whole city of Canton from the surface of the Pearl River, the church, as well as the European-style buildings in Thirteen Factories, constituted only a small fragment of the overall landscape of the city. The main urban landscape influence of the Anglican Thirteen Factories Church mainly lay along the river banks of the western suburbs, as the church mostly served both

westerners and water dwellers of lower social status who shuttled back and forth on the river (Figure 13).

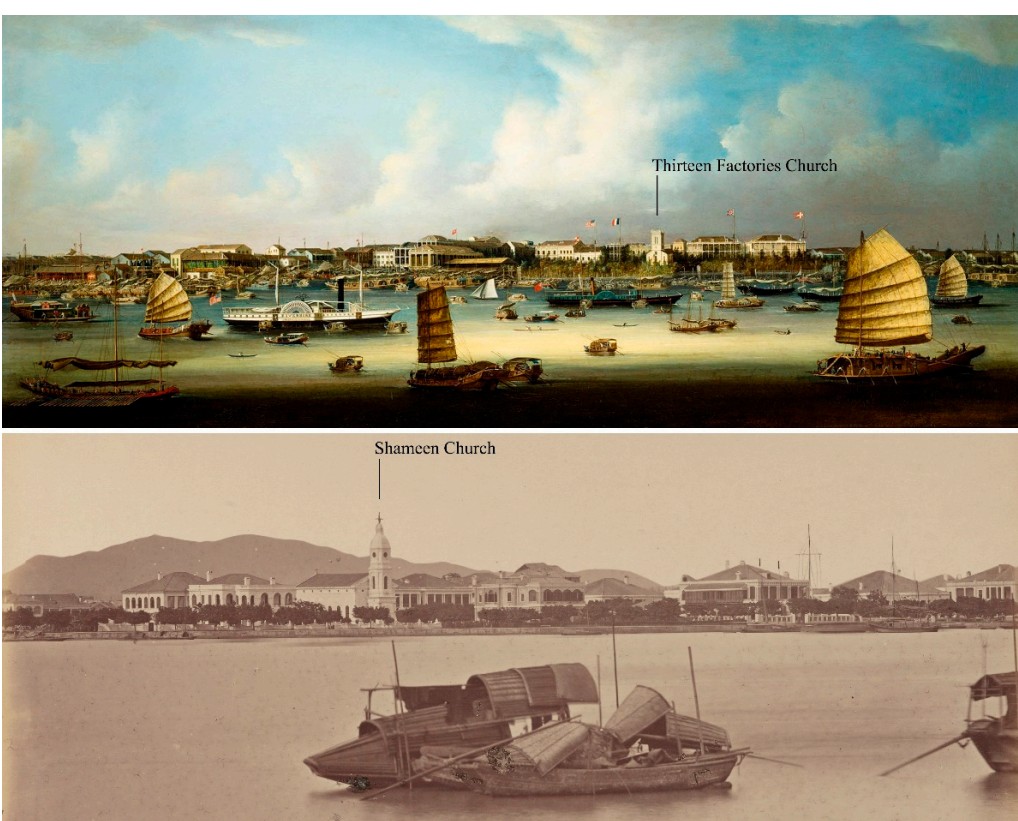

**Figure 13.** Landscape of the Anglican Thirteen Factories Church (about 1850–1855) and the Shameen Church (about 1870–1885, albumen silver print) on the north bank of the Pearl River. Source: (Hong Kong Museum of Art, and Peabody Essex Museum 1996, p. 197). https://www.getty.edu/art/collection/object/108ZRG (accessed on 30 July 2024).

The Anglican Shameen Church was also designed in the Italian style (Conner 2014, p. 231). The bell tower of the Shameen Church was about 28 m high, protruding from the main part with a cross standing on the dome. The church faced the Pearl River in a towering, complete, and independent posture. Looking from the Pearl River to Shameen Island, the Shameen Church was purer and more spectacular. However, the Shameen Church was still set against the background of other European-style buildings. From the perspective of the urban landscape, the buildings in Shameen Island were no different from the Thirteen Hongs and the Thirteen Factories Church, but on a larger scale, they became part of a more prominent colonial building complex along the riverside area of the western suburbs outside the city wall (Figure 13).

Compared to the two British Anglican churches, the Roman Catholic Cathedral of the Paris Foreign Missions Society caused a structural change in the urban landscape. It was incompatible with the local urban landscape of Canton, breaking the balance of *fengshui* and the original sense of order of the urban landscape, and was an obvious symbol of the anti-Confucian order (Wiest 2004, p. 243). In M. Guillemin's plans, this church should not have just met the needs of the Catholic community, it should have been built as grand and eye-catching as possible, leaving the first impression of Christianity on Chinese travelers in Canton. Compared with the pagodas and mosques in Canton, this church also became a monumental building, representing the glory of France (Masson 2019, pp. 116–18). Regarding the architectural style, Alfred Ducat, the chief architectural consultant of M. Guillemin, believed that a Byzantine or Gothic style would have been more suitable for

Canton's weather than the Greco-Roman style. After that, Alfred Ducat discussed with M. Guillemin about building a Byzantine church, but M. Guillemin finally opted for the Gothic style (Masson 2019, pp. 118–20). Judging from the export paintings depicting Canton at the end of the nineteenth century, the Roman Catholic Cathedral fulfilled M. Guillemin's wish (Conner 2014, pp. 224–26). Viewed from the surface of the Pearl River, it was no longer situated in a small corner but visually located in the center of the entire Canton. The buildings along the river and even the city wall south of the cathedral became the "base" of the Roman Catholic Cathedral. The Smooth Pagoda and the Flowery Pagoda were extremely small in comparison, and the Baiyun Mountain to the north of Canton served as its background (Figure 14). In a letter dated 27 July 1882, Bishop Augustin Chausse wrote that he saluted the beautiful spires of the cathedral as he was leaving Canton by boat on the Pearl River. These spires towered over other buildings in Canton just like the towers of Notre Dame or the Pantheon (L'Oeuvre 1883, p. 21).

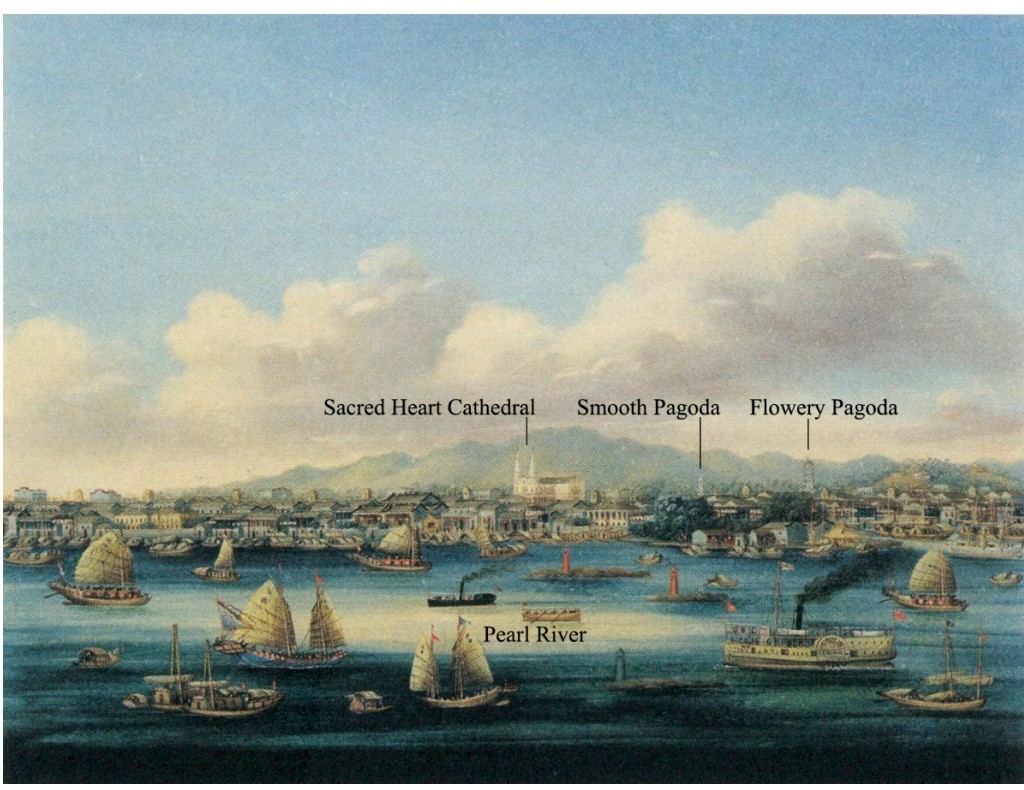

**Figure 14.** Painting—a full view of Canton seen from across the Pearl River (about 1885).[13] Source: (Conner 2014, p. 226).

Some Westerners who visited Canton and saw the Roman Catholic Cathedral praised the cathedral highly, but at the same time expressed concerns about the confrontation and hatred that the cathedral might arouse. Scottish travel writer Gordon Cumming believed that this majestic cathedral rose from the ground—it was a beautiful thing in itself, but it constantly reminded the citizens of Canton of the injustice and robbery of the invaders (Gordon Cumming 1887, pp. 43–44). During the riots of the early 1880s, the Cathedral was the focal point for the local inhabitants to vent their pent-up resentment and frustration. Most natives called it the "Fangui Stone Chamber",[14] expressing their contempt and deep hatred (Wiest 2004, pp. 231–52; Coomans 2019, p. 189). A. H. Exner, a German banker who visited Canton in 1886, conducted a more pertinent and appropriate evaluation. He believed that this symbol of Christian faith in a pagan environment would certainly leave an exciting impression on every Christian who saw it, but it was undeniable that such a large church building was too stark in contrast to the local temples and would inevitably arouse the anger of pagans, especially the hatred of missionaries. Exner also said that most

Europeans living in Canton believed that it was more desirable to build small churches in various locations, which they believed would achieve their purpose and arouse less hatred from the pagans (Exner 1889, p. 3).

Notre Dame de Lourdes, also in Gothic style, was located on Central Avenue, at the junction of the British concession and French concession (W. Li 1996, pp. 195–96). It was not planned along the river like the Anglican Shameen churches. Moreover, the bell tower of the church was only about 24 m high, and its external landscape declaration function was much weaker than that of the Anglican churches, while its internal services were more convenient.

The Southern Baptist Convention also adopted a Gothic style for the Dongshan Church, which was originally built in an underdeveloped area in the eastern suburbs. However, the Dongshan Church community facilitated interactions with urban development, creating positive conditions for the modernization of the urban landscape. The Dongshan area chosen by the Southern Baptist Convention still had a rural and suburban-style during the early Qing Dynasty, with low land use, large pastures, dense pine forests, and temples (Editorial Committee of Gazetteer of Dongshan District 1999, pp. 77–80). Compared to the New City and the Old City, the land price in the Dongshan area was lower and there was a large area of undeveloped land. The Southern Baptist Convention spent funds purchasing sufficient land and relocating the scattered churches and various facilities in the city to the Dongshan area. They also planned this area according to the Christian residential compound, the construction of which not only faced minimal social resistance but also involved the participation of the Chinese Baptist Church. The construction of the Dongshan Church community attracted an inflow of funds and people. Afterward, bureaucrats and expatriate Chinese also successively purchased land and built houses in Dongshan, gradually building large numbers of Western-style buildings and gardens (Figure 15). Dongshan became a gathering area for wealthy and noble people (Ye 2012, pp. 10–21). In the newly formed Western garden-style urban landscape, the Dongshan Church became a beautiful piece of scenery that complemented the entire neighborhood.

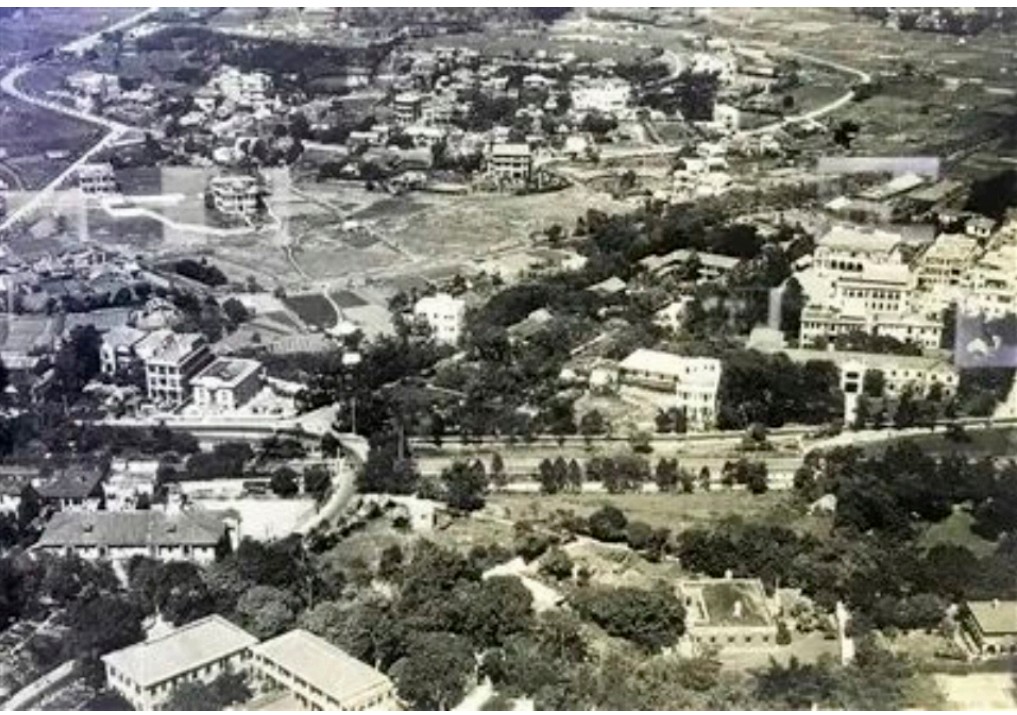

**Figure 15.** Photo of the Dongshan area in the early 20th century.[15] Source: (Zhang and Zeng 2022, A4 version).

In addition, gospel boats on the Pearl River could not be ignored. Gospel boats were basically made from wood with a simple appearance, but seemed larger than the boats of ordinary people on the river, and had little influence on the landscape of the Pearl River (Du 2009, pp. 66–68). However, missionaries set up schools and donated medical supplies on gospel ships, which improved the backward conditions of the residents who lived on the water (Yuxi Wu 2005, p. 33). The floating Bethel in the Whampoa district had six Gothic windows. Since the Whampoa district was an anchorage for foreign merchant ships, this bethel naturally increased the sense of stability for drifting Westerners (Bridgman 1850, p. 168). From the perspective of the landscape along the Pearl River, similar to the role of the *fengshui* pagodas in Huangpu, this development extended the Christian landscape to the estuary and enhanced the overall sense of sequence.

## 5. Conclusions and Discussion

On the one hand, churches, which occupied a central position in terms of functions and landscapes in Christian cities and parishes, conflicted with the rules of urban landscape construction in Canton. The city walls of Canton demarcated functional areas with different dominant functions and statuses, but at the same time, it was difficult to define a clear city center. The urban landscape in Canton presented a sense of balanced order that extended horizontally, lacking a visual center to dominate the entire city. Moreover, due to building height and width restrictions, the status of important buildings was mainly highlighted through the planar organization of cities and blocks. On the other hand, the differences in perspectives between the Qing government and the Western missionaries in viewing and envisioning Canton's urban landscape provided a buffer space for the expansion of churches. The Qing government paid more attention to the perspective from the north to the south of the city, while Western missionaries focused more on the route from the surface of the Pearl River to the north. This means that the churches entered urban neighborhoods without having a strong influence on Canton's official authority. Considering these two factors, the degree of conflict between the churches and the traditional urban landscape of Canton depended on the planning strategies and goals adopted by the missionaries.

Throughout the whole process, the expansion of Christian churches in Canton started from the banks of the Pearl River and then went northward to the main life veins within the city walls and the suburbs of Canton, which was in line with the activity trajectory of Westerners and was also conducive to missionary work. In the first stage (from 1582 to 1732), under the backdrop of government control over the scope of missionary work and church construction approvals, the entry of Catholic churches into Canton was pragmatic and cautious. Their site selection and construction reflected a strategy of integrating into the urban order. In the second stage (from 1844 to 1911), in addition to being used as a place to spread Christianity to local people or a place for Western Christians to conduct religious activities, churches were also used as symbols to declare the existence of Christians and Western forces (Coomans 2016, p. 192), which complicated the situation.

Since the core direction for the declaration faced the Pearl River, the layout planning and orientation of the church's bell tower, which bore the declaration function, were subject to this goal, but the general construction rules of western churches and parish centers, as well as the convenience of actual use, were ignored. In general, the influence of Protestant churches on the urban landscape was fragmentary rather than global. The fragments were connected by the main streets and alleys, forming an organized network and integrating into daily life. In contrast, the Roman Catholic Cathedral, as the symbol of the Holy See's power in Canton, towered over all buildings and changed the overall landscape pattern of the city, which made the entire city of Canton become the foreground and background of the cathedral. This broke the original sense of balance extending in the horizontal direction and profoundly changed the traditional logic of Canton's landscape order organization. It is worth mentioning that—in the case of Japan—the Franciscan missionaries also pursued

the verticality of their buildings in order to distinguish them from Buddhist temples and attract public attention (Arimura 2021, p. 203).

The missionary societies also built educational, medical, charitable, and publishing facilities to improve the living conditions of local people, especially those at the bottom of society. These facilities attracted Christians, forming a Christian community centered around the churches. Because of this dynamic, missionary societies competed with the government for spatial power, aiming to influence the urban functional system and spatial layout that had previously been dominated and controlled by the government through the planning and construction of the Christian community. The Paris Foreign Mission Society initiated the planning history of church communities and centers, but it built the Roman Catholic Cathedral on the site of the former Governor's Office in the New City. This choice resulted in limited development space and adopted the form of closed blocks to ensure safety, thus limiting its interaction with the city. In contrast, the Southern Baptist Convention built the Dongshan Church community in the undeveloped eastern suburbs, far away from government offices and commercial-intensive areas. On the one hand, the Southern Baptist Convention adopted an open-planning layout, and on the other hand, it led the Chinese churches and Chinese businessmen overseas to jointly develop the Dongshan area. This also proved the good relationship between the Baptist Convention and the local community. The Dongshan Church community promoted the planning and development of the eastern suburbs and even improved the political status of these suburbs.

After 1911, history shows that the church's value as a symbol of Canton's urban landscape was not long-lasting, but its function of integrating into culture and society continued. After the fall of the Qing Dynasty, the new Canton government formulated an urban planning scheme and built a number of modern buildings of mixed styles on the new embankment of the southern suburbs. Among these buildings, the tallest one, the Building of Da Sun Co. Ltd. (50 m high, 38 m wide from east to west, and 89 m long from north to south) became the most eye-catching star along the north bank of the Pearl River. In contrast, the Roman Catholic Cathedral gradually disappeared from view when viewed from the Pearl River (J. Wu 2018; Zou 2012). In comparison, the Dongshan Church still has social vitality, schools are still in use, and the Western-style building complex has become a historical and cultural protection area recognized by the government, as well as a highlight of cultural tourism (Zhang and Zeng 2022, A4 version).

Overall, from a historical perspective, the Christian churches built by Western missionaries in Canton underwent a process of emergence, prominence, and concealment in modern high-rise buildings. Correspondingly, the local attitude toward Christian churches evolved from initial contact to passive acceptance, and ultimately to diminishing influence. The different fates of Catholic and Protestant churches in Canton indicated that adaptive strategies for evangelism and church construction, as well as close interactions with local society and culture, contributed to the formation of diverse cultural landscapes, which were more conducive to the integration and coexistence of different cultures.

**Funding:** This research was funded by National Natural Science Foundation of China, grant number 51678362.

**Institutional Review Board Statement:** Not applicable.

**Informed Consent Statement:** Not applicable.

**Data Availability Statement:** The papers, books, documents, maps, and photos used in this article are all published or in the public domain. See the resource list and reference list for details.

**Acknowledgments:** I am very grateful to the anonymous reviewers for their insightful comments on this paper.

**Conflicts of Interest:** The author declares no conflicts of interest.

## Notes

1   In 1807, the London Missionary Society sent missionary Robert Morrison to China to preach, becoming the first Protestant missionary to enter Canton. At this time, the Qing government did not lift the maritime ban, and Protestant missionaries used the Thirteen Factories as a major center, engaging in covert missionary activities through serving in foreign commercial institutions in China, or running hospitals, schools, and publishing houses. However, there are no records of churches, so it is not within the scope of this study. (W. Li 1996, p. 234).

2   In the fourth year of the Shunzhi reign in the Qing Dynasty (1647), the Governor General presided over the construction of bird-wing city walls, about 7 m high and 5 m wide, at the east and west ends of the south of the New City to protect merchants along the river in the southern suburbs (Z. Zeng 1991, pp. 378–80).

3   Representations of the letter numbers in the right figure: a—the Pearl River; b—the foreign factories; c—Mobammedan mosque; d—a native pagoda (Flowery Pagoda); e—five-storied pagoda; f—the Governor's House (Viceroy Office, 总督衙门); g—the Foo-yuen's house (巡抚衙门); h—the house of Tseang-Keun or the Tartar General (将军衙门); i—the house of the Hoppo (the Superintendent Office of Customs, 河泊所); k—the house of the Heo-yuen (literary chancellor of Canton); l—house of the Poo-ching-sze (treasurer of the provincial revenue); m—the house of the Gan-cha-sze (criminal judge of the province); n—the house of the Yen-yun-sze (superintendent of the salt department); o—Kung-yuen (a hall for the reception of literary candidates at the regular examinations, 贡院); p—Yuh-ying-tang (a founding hospital); o—Teen-tsze ma-taou (the execution ground, 天字码头/刑场); The elements depicted in these two pictures are basically identical and should be of the same origin. However, the content of the English map is more simplified. Although there are no place names marked, the red houses, streets, city walls, bunkers, and river ditches on the map are easier to identify. These small red houses represent key facilities such as government offices, schools, temples, and defense. From this, it can be seen that these two pictures are essentially abstract expressions of these key facilities, city walls, and streets from the official perspective of the Qing court.

4   The drawer is standing on the rooftop of the British Trading House in Thirteen Factories. The upper picture is looking toward the south to the Pearl River, and the number 5 in the picture is the distant view of the Whampoa Pagoda. The lower picture is looking toward the north toward the western suburbs, the New and the Old cities, and the number 43 in the picture is the Five-Story Tower.

5   The photo of Venice was collected and digitally imaged by Heritage Auctions, HA.com. This panorama was mounted on a card in nine sections, and folded into a leather slipcase.

6   In 1742 (the seventh year of the Qianlong reign), due to the inability of the nursery houses outside the West Gate to meet the housing needs of women and children, the Canton authorities approved local merchants to build a new nursery in the Dongshan area outside of the East Gate. This new nursery was located in the southeast corner in Figure 2.

7   The locations of the churches are confirmed through argumentation in the article.

8   A—British churches and restaurant; B—the residence and the school of American Presbyterian; C—the residence and the chapel of Wesleyan Methodist Missionary Society; D—the Chinese Church of London Missionary Society; E—the house and the hospital of American Presbyterian; F—the residence, chapel, and school of Rhenish Missionary Society; G—the Roman Catholic Cathedral (under construction) and the orphanage; H—the residence and the church of Wesleyan Methodist Missionary Society. It should be noted that not all missionary societies had a church or a chapel.

9   For the English names and descriptions of the streets in Canton, please refer to (Kerr 1880, p. 25).

10   On 16 November 1853, the Holy See appointed M. Guillemin as the Apostolic Vicariate of Guangdong, Guangxi, and Hainan provinces. This appointment was announced to the public in 1855. Quoted in "Guangzhou Shengxin Dajiaotang de Sheji he Jianzao", written by Mattieu.

11   Jean-Paul Wiest highlighted in his research on the history of cathedral construction that "Bishop Guillemin in particular was politically, culturally, and religiously ill prepared to make himself accepted by the Chinese people. He epitomised the mentality so pervasive in the second half of nineteenth-century French Catholicism. He looked down upon almost every aspect of Chinese culture and regarded himself as a pioneer of Catholicism and French civilisation. In his mind, the two were inseparable and his cathedral in Guangzhou was to be the sign of the implantation of both on Chinese soil. Through persuasion, ruse, and diplomacy, he manipulated French civil and military authorities both at home and in China to assist willy-nilly in his project." (Wiest 2004, p. 250).

12   1a—the Roman Catholic Cathedral; 1b—Bishop's Administrative Management Center; 1c—seminary; 1d—male school or Roman Catholic Academy; 1e—church houses; 2a—female school; 2b—orphanage; 2c—craft factory; 2d—church houses.

13   The tall building with two bell towers in the middle of the screen was the Roman Catholic Cathedral, and the two tower-shaped buildings on the right were the Flowery Pagoda and the Smooth Pagoda.

14   A stone chamber, on the surface, means that a church is a house built with stones. In the 19th century, most buildings in Guangzhou used bricks, soil, and wood materials. Quoted in the summary of the building materials that Westerners saw in Guangzhou, In *Xifangren Suzao de Guangzhou Jingguan*, written by Ni (2007, pp. 53–54).

15   The photo shows the north side of the Dongshan Church. The east–west road in the lower middle is the Guangzhou-Kowloon Railway.

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
