# Peer review of "Spatial Expansion, Planning, and Their Influences on the Urban Landscape of Christian Churches in Canton (1582–1732 and 1844–1911)"

_religions, doi:10.3390/rel15101183_

Round 1

Reviewer 1 Report

Comments and Suggestions for Authors

Your effort is much appreciated, particularly for demonstrating the data and figures of the architecture and buildings. Visual materials like maps and photos help the reader grasp the spatial dimension of Canton in contexts.
Some suggestions may be helpful. Firstly, the overall discussion can be more in-depth. I enjoyed reading the conclusion, as it shows a sophisticated discussion of interaction, conflicts, dynamics, and tensions between parties and forces due to political and societal influences. The study contains many facts and figures that show your effort. Still, the portion discussing and analyzing the underlying factors, such as conflicts and negotiation among different parties, may not be enough. Secondly, a lack of a theoretical framework makes the study less sophisticated. Orientalism, "assimilation and indirect," and the relational approach are some of the examples. Some other Western literature, particularly those studies on the conflict between religious agencies, political authorities, and other religious agencies, provide concrete frameworks. 
Thirdly, you seem to mix up the "reference list" and the "footnotes/ endnotes."

This study is a valuable resource for understanding the development of Christianity in China, particularly in Canton.

Author Response

Reviewer 1

Your effort is much appreciated, particularly for demonstrating the data and figures of the architecture and buildings. Visual materials like maps and photos help the reader grasp the spatial dimension of Canton in contexts.

Some suggestions may be helpful.

Comment 1: Firstly, the overall discussion can be more in-depth. I enjoyed reading the conclusion, as it shows a sophisticated discussion of interaction, conflicts, dynamics, and tensions between parties and forces due to political and societal influences. The study contains many facts and figures that show your effort. Still, the portion discussing and analyzing the underlying factors, such as conflicts and negotiation among different parties, may not be enough.

Response 1: I would like to express my gratitude for the reviewer’s generous suggestions and helpful comments. After taking all the comments into account, the structure of the article was reorganized in order to explore the research content in a detailed and in-depth manner. Combining with the second comment, I draw on the theoretical ideas of modernization paradigms and postcolonial theory. Based on the expansion of the churches themselves and the evolution of their relationship with Canton, this article evaluates the impacts of missionary work on the urban landscape order of Canton, reveals the interactive relationship between Catholicism/Protestantism and Canton society in different periods, and analyses the church construction strategies of Catholicism and Protestantism which constitute a part of the history of missionary work.

Comment 2: Secondly, a lack of a theoretical framework makes the study less sophisticated. Orientalism, "assimilation and indirect," and the relational approach are some of the examples. Some other Western literature, particularly those studies on the conflict between religious agencies, political authorities, and other religious agencies, provide concrete frameworks.

Response 2: I appreciate the insightful and constructive comments. This prompts me to re-examine the previously submitted article and calm down to seek theoretical support for my research. I have read relevant historical and theoretical literatures on missionary works, including the research findings of Ryan Dunch, Wang Lixin, Jean & John Comaroff, Edward W. Said, Paul W. Harris, Arif Dirlik, Roger R. Thompson, and others, which drives me to recognize the academic reflection on theories such as "Cultural Aggression", "Cultural Imperialism", "Orientalism", and "Cultural Imperialism”.

Drawing on Wang Lixin's view, "the modern Christian missionary movement in China was an extremely complex historical phenomenon. This complexity lied in the extensiveness of missionary activities, the multiplicity of their social relations and the mult-level impacts". "Perceiving the modern missionary movement in China through the modernization paradigms and postcolonial theory may be closer to the historical truth." As an objective existence, churches can serve as an appropriate carrier for observing and evaluating the history of missionary work. With the help of the modernization paradigms and postcolonial theory, this paper adopts the cross-method of architecture, history and sociology to analyze the expansion of churches in Canton by the missionary society. By means of historical documents and images, it analyzes the intentions and objective influences of missionaries, the attitudes of Canton residents, and the interactive relationship between missionaries and Canton residents.

It is undeniable that this article focuses on historical empirical research and still cannot reach the level of classical research in terms of theoretical discussion. However, combined with the research on the history of Christian missionary work in China, the academic contribution of this article lies in analyzing and evaluating the differences in church construction strategies and their impacts by combing the historical process of the churches’ construction by Catholic and Protestant missionary societies. This provides new content and evidence for the study of missionary history and, to a certain extent, builds an interdisciplinary research approach.

Thanks again to the reviewers for motivating me. I will continue to work hard to promote more in-depth research.

Comment 3: Thirdly, you seem to mix up the "reference list" and the "footnotes/ endnotes."

Response 3: My sincere thanks to the reviewer for pointing out the citation problem. According to the reviewer's comment, I reorganized the References and the Notes with reference to the format of the articles published under the theme of "Chinese Christianity: From Society to Culture."

Once again, I would like to express my gratitude for these constructive and insightful comments, which greatly helps me improve the article.

Reviewer 2 Report

Comments and Suggestions for Authors

I read this article with great interests. It examines the urban landscape of Guangzhou under the influence of construction of Christian churches, which is an interdisciplinary study of architecture, sociology, and history. Impressively, the author provides many figures and pictures of the development of Christian churches in Guangzhou. However, this article lacks a comprehensive analysis of the Christian churches as a whole, and there are some logical problems that need to be fixed before publication. 

1. The author's choice of time span in the title is unreasonable, and the article does not give a convincing reason.

The title states that the author wants to analyze the changes in the urban landscape between 1581 and 1911, but when the author further explains the choice of time span in the article, he/she simply says that the situation during this period was "ignored". This period is actually quite long, which covers more than 100 years. The time span in the title is roughly 300 years, if one third of the span is omitted,  it indicates the illogicality of the time scope in the title. It is not reasonable to cut such a long time out of the study.

2. The article lacks a profound review of the current state of scholarship about Guangzhou and Christian churches. 

In the introduction, the author only focuses on the history of Christian missionaries, while fails to give the whole picture of the studies about Guangzhou's landscape of Christian churches. Other studies of cultures and church history should be included and referred to. The author needs to point out clearly what makes this study stand out and contribute to the field, especially to the theme of this special issue, ‘Chinese Christianity: From Society to Culture.’

There are other parts that should include more references. For example, when the author talks about the spatial layout of Guangzhou (line 72-80), the readers could not find any sources of historical data in this part. It seems that the author describes the background merely from his/her own perspective. 

3. Some of the conclusions of the article are not grounded, so the conclusions seem a bit subjective. 

For example, line 458-459, the author says that "In the early Qing Dynasty, although the strong power of the Qing court restricted missionaries, this restriction actually prompted the missionaries to build churches in a targeted manner", but the "restrictions" are not mentioned in the whole article. It only mentions how Francesco Sambiasi broke the restrictions of the city wall, no other governmental restrictions  could be found in the whole text. The author needs to give more evidences to make the conclusions convincing.

Besides, the author only selects "several churches that have been preserved" (line 67), the process of selection lacks a valid reason.It is arbitrary to select the churches as cases for study based on their preservations.  And the author also needs to make it clear that the selected churches are representative for the study. Now the churches chosen here seem to be a quite fragmentary set of examples.

And from line 220-254, the author spent a lot of space to list historical facts, but in the end he came to a conclusion that was self-evident: “ If this was the case, it would mean that after more than half a century of operation, the missionaries had a deeper understanding of the spatial layout and functional facilities in Canton. Without contradicting official architectural authority, churches proactively integrated into key public spaces within the city”.(line 261-263) The logical connection between the historical facts and the conclusion is not clear.

4. Some of the figures and pictures used in the study are not compatible with the textual analysis. 

For example, the author describes figure 5 on page 6 as "Map of Guangdong Province in the late 18th Century". However, I don't think this picture is a map. The Chinese characters in this picture is not clear enough to convey any information. The readers could only see some boats clearly. 

Another issue is about the references of the pictures and figures. In the references, most of the author's pictures come from the Internet, and some are from popular networks such as Wikipedia, rather than academic databases. The author should find pictures in historical archives or electronic archives and then cite them in the paper, so that the entire research can be more academic.

5. The entire article should be restructured and its content should be further enriched.

For example, there is no transition paragraph between 3 and 3.1, neither between 4 and 4.1; 4.4 is extremely short and the section is not proportional.

Comments on the Quality of English Language

The language of this article is not academic enough, and many of the English expressions are very colloquial, so it is recommended that the author further revise and polish the English expressions.

Author Response

Reviewer 2

I read this article with great interests. It examines the urban landscape of Guangzhou under the influence of construction of Christian churches, which is an interdisciplinary study of architecture, sociology, and history. Impressively, the author provides many figures and pictures of the development of Christian churches in Guangzhou. However, this article lacks a comprehensive analysis of the Christian churches as a whole, and there are some logical problems that need to be fixed before publication.

Comment 1: The author's choice of time span in the title is unreasonable, and the article does not give a convincing reason.

The title states that the author wants to analyze the changes in the urban landscape between 1581 and 1911, but when the author further explains the choice of time span in the article, he/she simply says that the situation during this period was "ignored". This period is actually quite long, which covers more than 100 years. The time span in the title is roughly 300 years, if one third of the span is omitted, it indicates the illogicality of the time scope in the title. It is not reasonable to cut such a long time out of the study.

Response 1: I owe a special debt of gratitude to the reviewer for this issue. In the seventh paragraph of ‘Introduction’ of the revised manuscript, a detailed explanation was provided as to why it was divided into two stages: 1582-1732 and 1844-1911 (line**). First, the year 1581 was corrected to 1582. The reference for 1581 is ‘Henri, Bernard. Tianzhujiao Shiliu Shiji Zaihua Chuanjiaozhi [Missionary Records of Catholicism in China in the 16th Century], Beijing: The Commercial Press, 1936: 190-191.’, and the reference for 1582 is ‘Pfister, Louis S. J.. Notices biographiques et bibliographiques sur les J ésuites de l'ancienne mission de Chine (1552-1773), Tome I. Chang-hai (China): Imprimeriede la mission catholique. Orphelinat de Tou-sé-wé. 1932: 136-142.’. Referring to the second French literature should be more appropriate. Second, local gazetteers record that the Canton authorities demolished the churches in Canton in 1732 and expelled missionaries to Macau. Until 1844 when the Treaty of Wanghia was signed by Sino-the United States and the Treaty of Whampoa signed by Sino-France, these two treaties stipulated that the American and the French had the right to rent land and build churches in five trading ports. There is no record of churches from 1733 to 1843 in the Religious Annals of Canton. Therefore, for the sake of a comparative study, this article ignores the situation from 1733 to 1843 as there is no historical data on churches at hand and classifies the period from 1844 to 1911 as the second stage.

In addition, although some missionaries operated secretly from 1733 to 1843, especially those who acted as foreign merchants. However, due to the strict prohibition of missionary policies by the Qing government, there was indeed no evidence of churches existing in Canton. If the time span of the research is inappropriate, may I ask if it is possible to divide it into two periods (1582-1732, 1844-1911) as explained above?

Comment 2: The article lacks a profound review of the current state of scholarship about Guangzhou and Christian churches.

In the introduction, the author only focuses on the history of Christian missionaries, while fails to give the whole picture of the studies about Guangzhou's landscape of Christian churches. Other studies of cultures and church history should be included and referred to. The author needs to point out clearly what makes this study stand out and contribute to the field, especially to the theme of this special issue, ‘Chinese Christianity: From Society to Culture.’

There are other parts that should include more references. For example, when the author talks about the spatial layout of Guangzhou (line 72-80), the readers could not find any sources of historical data in this part. It seems that the author describes the background merely from his/her own perspective.

Response 2: My warm thanks to the reviewer for pointing out the issues. Previous studies have focused on the design and construction of churches, showcased the details of conflicts or collaborations between missionary churches, Canton officials, and the public during the construction of the churches, and analyzed the architectural characteristics of churches in Canton. In addition, other studies have also emphasized on the interaction between local elite groups, residents and foreigners in Canton's urban planning and construction under the influence of foreign forces. Based on these achievements, this article aims to study the spatial expansion process of Christian churches as an urban landscape element in Canton after Christianity was introduced into this city, as well as their impacts on the city’s urban landscape during this process. On the one hand, the church was a core element of traditional cities with Christian beliefs in the West, often occupying a central position in Christian cities or communities; On the other hand, for the cities under the Confucian order in China, churches were very different heterogeneous elements, especially those with bell towers, which would inevitably bring an impact or even a shock on the traditional Chinese urban landscape order that emphasized imperial power. Compared to the existing researches, this article further examines the expansion of churches and their interactions with the city from the perspective of the overall city, providing a new multidisciplinary approach to analyzing and evaluating the history of missionaries and churches construction (line 29-59).

Therefore, in terms of content, this article is related to the three aspects mentioned in themes: “In the Special Issue, original research articles and reviews are welcome. Research areas may include (but are not limited to) the following: 1. Jesuits’ role in the transition of Late Ming and Early Qing culture and society; 2. Protestant missionaries’ activities and their effects on national, regional and local levels since the Late Qing period to modern day; 3. Christian communities and their relations with local society and culture.”

Thank the reviewer again for pointing out the issue of the references. The revised manuscript has removed the line 72-80 from the original manuscript. The analysis of the spatial layout and other content of Canton is relied on historical documents and images to avoid the problem of subjective description.

Comment 3: Some of the conclusions of the article are not grounded, so the conclusions seem a bit subjective.

For example, line 458-459, the author says that "In the early Qing Dynasty, although the strong power of the Qing court restricted missionaries, this restriction actually prompted the missionaries to build churches in a targeted manner", but the "restrictions" are not mentioned in the whole article. It only mentions how Francesco Sambiasi broke the restrictions of the city wall, no other governmental restrictions could be found in the whole text. The author needs to give more evidences to make the conclusions convincing.

Besides, the author only selects "several churches that have been preserved" (line 67), the process of selection lacks a valid reason.It is arbitrary to select the churches as cases for study based on their preservations. And the author also needs to make it clear that the selected churches are representative for the study. Now the churches chosen here seem to be a quite fragmentary set of examples.

And from line 220-254, the author spent a lot of space to list historical facts, but in the end he came to a conclusion that was self-evident: “ If this was the case, it would mean that after more than half a century of operation, the missionaries had a deeper understanding of the spatial layout and functional facilities in Canton. Without contradicting official architectural authority, churches proactively integrated into key public spaces within the city”.(line 261-263) The logical connection between the historical facts and the conclusion is not clear.

Response 3: I am grateful to the reviewer for pointing out the problem. I have added some literature support to avoid subjective induction and kept a more cautious attitude in drawing conclusions.

First, regarding the issue of "restrictions on missionary work", I consulted relevant literature on the history of missionary work in Canton. In the Introduction section, I explained that “during the late Ming and early Qing dynasties, western missionaries carried out missionary work and built churches with the permission of the emperor or local administrative officials. Conversely, missionaries also actively established good relationships with the emperor and the local officials, for obtaining the official permission and protection to facilitate missionary work.” And the revised draft also removed inappropriate conclusions from the original manuscript.

Second, the revised manuscript explains the issue of case selection in the Introduction section. “Regarding the differences in service targets and construction purposes, the division between Catholicism and Protestantism, and the diversities in spatial composition and organization, this article selects the following typical examples: the Thirteen Factories Church and the Shameen Church established by British Anglican, the Roman Catholic Cathedral (also known as the Sacred Heart Cathedral) by Paris Foreign Missions Society and the Dongshan Church by Southern Baptist Convention.” (line 99-105). Such choice takes into account the theoretical frameworks of "cultural imperialism", "modernist paradigms", and "postcolonial theory”. Although the uniqueness of these four examples was previously discovered, it was not explained from an academic perspective. The Thirteen Factories Church and the Shameen Church mainly served westerners and were located in the enclosed concession area. The Roman Catholic Cathedral of the Paris Foreign Missions Society was used as a symbol to declare the existence of Western power and Christian faith. The Dongshan Church and its community were successful examples of good interaction between the Southern Baptist Convention, local Chinese churches and the chambers of commerce. As pointed out in the Conclusion and Discussion of the article, the Dongshan Church was more closely integrated with the urban context and society of Canton than the Roman Catholic Cathedral was (line 680-696). There are relevant images and detailed information about churches, which is actually a side proof of their representative status.

Third, inappropriate inferences were removed from Chapter 3. In Chapter 3, a significant amount of space was devoted to verifying the locations of seven churches in 1704 and implementing them in the local chronicles map, as previous research had not done this work, which itself made a certain contribution. The value of verifying the location of churches lies in sorting out the scope of missionary activities in Canton during the late Qing and early Ming dynasties (1582-1732), and then summarizing the relationship between the missionaries, Canton officials and local residents by examining the distribution of churches and their related activities. This can also reveal the strategies adopted by missionaries in building churches and preaching.

Comment 4: Some of the figures and pictures used in the study are not compatible with the textual analysis.

For example, the author describes figure 5 on page 6 as "Map of Guangdong Province in the late 18th Century". However, I don't think this picture is a map. The Chinese characters in this picture is not clear enough to convey any information. The readers could only see some boats clearly.

Another issue is about the references of the pictures and figures. In the references, most of the author's pictures come from the Internet, and some are from popular networks such as Wikipedia, rather than academic databases. The author should find pictures in historical archives or electronic archives and then cite them in the paper, so that the entire research can be more academic.

Response 4: I would like to thank the reviewer for careful review.

First, the revised manuscript removed the figure of “Map of Guangdong Province in the late 18th Century” and replaced it with a more suitable hand drawn map of “Figure 3 Description of a View of Canton 1838” to describe the landscape of Canton. The revised manuscript also replaced other inappropriate figures, and provided analysis and explanations in a corresponding part.

Second, in the revised manuscript, the photo sources on Wikipedia have been replaced with historical photos collected from published literature, government websites, or libraries. These historical photos and paintings are in a single form, being scattered in various libraries or art collections that can be found in academic databases. Unfortunately, there is still a photo of Venice from the 1870s (Figure 13) that could not be found as an academic source and is still available on Wikimedia Commons. Judging from the content, this photo should have been taken in person. It is feasible to compare and illustrate the urban landscape characteristics of Venice and Canton in this article.

Comment 5: The entire article should be restructured and its content should be further enriched.

For example, there is no transition paragraph between 3 and 3.1, neither between 4 and 4.1; 4.4 is extremely short and the section is not proportional.

Response 5: I really appreciate this meaningful comment. Reflecting on the comments, I re-examined the article to make sure that the core content discussed in the article includes three aspects: the spatial distribution expansion of the churches, the evolution of layout planning and their influences on Canton’s urban landscape.

The second chapter (The spatial layout and the landscape characteristics of Canton) are delineated from three aspects, namely, urban functional zoning and spatial layout, landscape features with horizontal extension, and the construction of Canton’s space order. The fourth chapter (The planning and spatial expansion of churches and their influences on urban landscapes under the weak situation of the Qing dynasty, 1844-1911) is also re-organized. Dividing the analysis of Christian churches after the Opium War into three parts: the expansion of churches from the Pearl River to the New city, the Old city and the suburbs, the layout planning progress of the churches, and the influences of the churches on urban landscapes. Such division corresponds to the structure of the second chapter. The new structure of this article helps conduct a clearer and more in-depth study.

In addition, a transition paragraph (line 339-344) has been added to Chapter 4. The various parts of each chapter are well balanced in narration.

Once again, I would like to express my gratitude for these constructive and insightful comments, which greatly helps me improve the article.

Reviewer 3 Report

Comments and Suggestions for Authors

The article presents an innovative and insightful approach by situating the historical process of cultural exchange between China and the West within the context of urban planning case studies. The selection of Guangzhou as the case study is a valuable choice. However, there are a few areas that could benefit from further improvement:

1. The language still requires further refinement. More careful proofreading and editing would help enhance the overall fluency and clarity of the writing.

2. The historical terminology should be more rigorous. For instance, the use of "missionary" is rather ambiguous; it would be better to emphasize specific figures such as Michele Ruggieri, a prominent Jesuit missionary. Additionally, there is no need to consistently include the Chinese emperor's reign titles when referencing historical periods, as it can appear overly cumbersome.

3. The number of images could be reduced, as some do not appear to have a strong connection to the textual content. A more selective and purposeful use of visuals would strengthen the overall coherence of the article.

4. The segment from lines 199-213, which attempts to discuss the relationship between pagodas and feng shui, lacks a clear and coherent argumentation. The introduction of the feng shui concept is not fully developed, and the connections made could be strengthened.

5. The section "4.4. Churches on the rivers with Canton characteristics" is not thoroughly discussed and could benefit from more in-depth analysis. Expanding on this section would provide readers with a more comprehensive understanding of the topic.

6. If the focus is on the cultural exchange between China and the West in the context of urban planning, the author could consider exploring the historical changes in European architectural styles and patron systems over an extended period, and how these might be associated with the development described in the article. This could potentially offer additional insights and depth to the analysis.

Overall, the article presents some contribution by examining the intersection of cultural exchange and urban planning, but the incorporation of these suggested revisions could further enhance the quality and impact of the work.

Comments on the Quality of English Language

The language still requires further refinement to polish away any residual traces of machine translation. More careful proofreading and editing would help enhance the overall fluency and clarity of the writing.

Author Response

The article presents an innovative and insightful approach by situating the historical process of cultural exchange between China and the West within the context of urban planning case studies. The selection of Guangzhou as the case study is a valuable choice. However, there are a few areas that could benefit from further improvement:

Comment 1: The language still requires further refinement. More careful proofreading and editing would help enhance the overall fluency and clarity of the writing.

Response 1: I feel grateful for pointing out language problem. According to the review comments, I carefully proofread and edited the entire text, improving the structure, sentences, and vocabulary to make the language more academic.

Comment 2: The historical terminology should be more rigorous. For instance, the use of "missionary" is rather ambiguous; it would be better to emphasize specific figures such as Michele Ruggieri, a prominent Jesuit missionary. Additionally, there is no need to consistently include the Chinese emperor's reign titles when referencing historical periods, as it can appear overly cumbersome.

Response 2: Thanks for the kind comment on the unclear semantic expression. I searched for the literature and reconfirmed the names of missionaries to ensure accurate historical information. Meanwhile, following the review comments, the political titles of the Chinese emperors were omitted. Additionally, the following is a brief table of the churches covered in the article.

Churches

Missionary societies

Country

Year

A chapel

Jesuit

Italy

1582

Kan-wouan-lei

Jesuit

Portugal

1704

Yoc-yu-tong

Jesuit

France

1704

Sin-keou-lei

One of them belonged to Augustine;

Two of them belonged to Paris Foreign Missions Society;

Two of them belonged to Franciscan

1704

Tsoc-tong-yamoun

1704

Tsim-lo-kong-koun

1704

Tai-toc-hang

1704

Tai-fat-se

1704

Temple of the true god

London Missionary Society

British

1845

Dongshijiao Church

Southern Baptist Convention

American

1845

Pwanting Street Church

Southern Baptist Convention

American

1845

Zidong-ting

Southern Baptist Convention

American

1846

Thirteen Factories Church

Anglican

British

1847

Thirteenth Factories Street Chapel

Paris Foreign Missions Society

France

1848

The floating bethel at Whampoa

---

---

1850s

Shameen Church

Anglican

British

1865

A chapel

Wesleyan Methodist Missionary Society

British

Before 1878

A church

Wesleyan Methodist Missionary Society

British

Before 1878

A chapel

Rhenish Missionary Society

German

Before 1878

Roman Catholic Cathedral

Paris Foreign Missions Society

France

1888

Notre Darme de Lourdes

Paris Foreign Missions Society

France

1893

Morning Star

Swedish Evangelical Free Church

American

1896

Dongshan Church

Southern Baptist Convention

American

1906

Notes: In addition, there were fifteen Protestant churches or chapels in Figure 9 “Map of Canton with churches and government offices marked (about 1880)”, eleven of which were constructed after 1878. However, due to limited materials collected at present, the corresponding missionary societies have not been confirmed and therefore are not listed in this table.

Comment 3: The number of images could be reduced, as some do not appear to have a strong connection to the textual content. A more selective and purposeful use of visuals would strengthen the overall coherence of the article.

Response 3: I would like to thank the reviewer for careful review. According to the review comments, I have removed unnecessary images and reorganized the retained images based on the new article structure. Meanwhile, based on the new article content, images have been added regarding the landscape and spatial organization characteristics of Canton, the distribution of churches, and the planning of Christian communities.

Comment 4: The segment from lines 199-213, which attempts to discuss the relationship between pagodas and feng shui, lacks a clear and coherent argumentation. The introduction of the feng shui concept is not fully developed, and the connections made could be strengthened.

Response 4: I am grateful to the reviewer for pointing out the problem. After considering the opinions of the three reviewers, I have reorganized this part of the content. In terms of the objective of the article, the discussion on the relationship between towers and fengshui deviates from the main line of the article. So, I changed the narrative direction of this part. In the revised manuscript, these towers were simply taken as the product of Fengshui, and the focus was that the towers along the Pearl River further expanded the landscape characteristics of Canton in the horizontal direction, which was totally different from the development of Christian cities in the Middle Ages in pursuit of vertical direction (line 187-230).

Comment 5: The section "4.4. Churches on the rivers with Canton characteristics" is not thoroughly discussed and could benefit from more in-depth analysis. Expanding on this section would provide readers with a more comprehensive understanding of the topic.

Response 5: I appreciate for the insightful and constructive comment. In the initial writing process, I found that the "Gospel Boat" on the Pearl River was an object worthy of special attention, so I listed it in a separate section. After searching for historical documents on evangelism again, unfortunately, there are not many materials available to support a single section discussion. Therefore, the revised manuscript places the content of the Gospel Boat in the corresponding sections of "4.1" (line 423-426) and "4.3" (line 631-640), and supplements some historical information and discourse based on the collected literature.

Comment 6: If the focus is on the cultural exchange between China and the West in the context of urban planning, the author could consider exploring the historical changes in European architectural styles and patron systems over an extended period, and how these might be associated with the development described in the article. This could potentially offer additional insights and depth to the analysis.

Response 6: I really appreciate this meaningful comment. According to the review comments, I have reviewed the existing researches on the history of churches construction in Canton and supplemented information on architectural style and the patrons, especially by matching each church with the corresponding missionary society and missionary (see "response 2"). These new details enrich the research conclusions and make them more profound.

Once again, I would like to express my gratitude for these constructive and insightful comments, which greatly helps me improve the article.

Round 2

Reviewer 2 Report

Comments and Suggestions for Authors

I can see that the author has made a lot of improvements in the second version of the article.  There are some minor suggestions for further revision:

1. The author mentioned that there is still a photo of Venice from the 1870s (Figure 13) that could not be found from academic sources. It would be better of there could be more information about the photographer, since the author mentioned it was taken in person.

2. In the last part of the article, "conclusion and discussion", the author focuses on the narration of the facts, such as the status of each church in Guangzhou right now. It would be more compelling if the author could give a general conclusion based on all the historical facts presented in the whole article.

3. As for the time span in the title, I personally prefer the second solution: divide it into two periods (1582-1732, 1844-1911) . However, I know it is not common in academic research to have two time spans in the title. The author could also ask the other reviewers for further suggestions.

Comments on the Quality of English Language

The quality of English language is much better than the first version. 

Author Response

I can see that the author has made a lot of improvements in the second version of the article. There are some minor suggestions for further revision:

Comment 1: The author mentioned that there is still a photo of Venice from the 1870s (Figure 13) that could not be found from academic sources. It would be better of there could be more information about the photographer, since the author mentioned it was taken in person.

Response 1: I am grateful to the reviewer for pointing out the problem. In the previous version of the response, I accidentally confused the figure numbers. I marked Figure 13, but it was actually Figure 4. In the newly revised version, relative information of 'the photo of Venice' (Figure 4) is added: "The photo of Venice was collected and digital imaged by Heritage Auctions, HA.com. This panorama was mounted on card in nine sections, folding in to leather slip case" (Line 217 and Note 6). I carefully searched about this photo but in vain still cannot find the original photographer. The original photos were collected by America's Auction House in Heritage Auctions and sold out on Nov.13, 2015. This photo shows the perspective of Venice from the harbor, with numerous churches protruding from the skyline, especially the St Mark's Campanile (the bell tower of St Mark's Basilica in Venice) occupying the visual center. This photo creates a vivid and strong contrast with the horizontally extended landscape features presented in the photos of Canton of the same period. I also tried to seek another one from the same period on academic databases to replace this one, but still cannot find an appropriate panorama photo.

Comment 2: In the last part of the article, "conclusion and discussion", the author focuses on the narration of the facts, such as the status of each church in Guangzhou right now. It would be more compelling if the author could give a general conclusion based on all the historical facts presented in the whole article.

Response 2: I really appreciate for the insightful and constructive comment. I made a conclusion from the historic perspective as the last paragraph. That is, “Overall, from a historical perspective, the Christian churches built by western missionaries in Canton have gone through a process of emergence, prominence, and concealment in modern high-rise buildings. Correspondingly, the local attitude towards the Christian churches has undergone a change from contact, to passive acceptance, and then to reducing influence. The different fates of Catholic and Protestant churches in Canton indicate that adaptive strategies for evangelism and church construction, as well as good interaction with local society and culture, contribute to the formation of diverse cultural landscapes, which are more conducive to the integration and coexistence of different cultures” (line 709-716).

Comment 3: As for the time span in the title, I personally prefer the second solution: divide it into two periods (1582-1732, 1844-1911). However, I know it is not common in academic research to have two time spans in the title. The author could also ask the other reviewers for further suggestions.

Response 3: I would like to express my gratitude for the reviewer’s generous suggestions. According to the review suggestions, I have decided to divide into two periods (1582-1732 and 1844-1911), as this is consistent with historical circumstances. Meanwhile, other reviewers did not raise any disagreement or comment on this time span. Although using two periods in the title is not common, it would be acceptable.

Reviewer 3 Report

Comments and Suggestions for Authors

The author has made substantial efforts in revising the paper, thoroughly incorporating the suggestions provided. The paper has been carefully polished based on the previous feedback. These efforts are commendable and add significant value to the work.

Comments on the Quality of English Language

The English has improved significantly compared to the previous submission, though some minor revisions are still needed.

Author Response

Reviewer 3

Comment 1: The author has made substantial efforts in revising the paper, thoroughly incorporating the suggestions provided. The paper has been carefully polished based on the previous feedback. These efforts are commendable and add significant value to the work.

Response 1: Thanks so much for the recognition and encouragement. Once again, I would like to express my gratitude for the constructive and insightful comments raised before.

Comments on the Quality of English Language: The English has improved significantly compared to the previous submission, though some minor revisions are still needed.

Response: I am grateful to the reviewer for pointing out the English language problem. I carefully checked the words, grammar, and semantic expressions again to avoid language problems. The revised content has been highlighted in the newly revised draft.